# INTERPRETABILITY WITH FULL COMPLEXITY BY CONSTRAINING FEATURE INFORMATION

Kieran A. Murphy[1] and Dani S. Bassett[1,2,3,4,5,6,7]

[1]*Dept. of Bioengineering, School of Engineering & Applied Science,*
*U. of Pennsylvania, Philadelphia, PA 19104, USA*
[2]*Dept. of Electrical & Systems Engineering, School of Engineering & Applied Science,*
*U. of Pennsylvania, Philadelphia, PA 19104, USA*
[3]*Dept. of Neurology, Perelman School of Medicine, U. of Pennsylvania, Philadelphia, PA 19104, USA*
[4]*Dept. of Psychiatry, Perelman School of Medicine, U. of Pennsylvania, Philadelphia, PA 19104, USA*
[5]*Dept. of Physics & Astronomy, College of Arts & Sciences, U. of Pennsylvania, Philadelphia, PA 19104, USA*
[6]*The Santa Fe Institute, Santa Fe, NM 87501, USA*
[7]*To whom correspondence should be addressed: dsb@seas.upenn.edu*

## ABSTRACT

Interpretability is a pressing issue for machine learning. Common approaches to interpretable machine learning constrain interactions between features of the input, sacrificing model complexity in order to render more comprehensible the effects of those features on the model's output. We approach interpretability from a new angle: constrain the information about the features without restricting the complexity of the model. We use the Distributed Information Bottleneck to optimally compress each feature so as to maximally preserve information about the output. The learned information allocation, by feature and by feature value, provides rich opportunities for interpretation, particularly in problems with many features and complex feature interactions. The central object of analysis is not a single trained model, but rather a spectrum of models serving as approximations that leverage variable amounts of information about the inputs. Information is allocated to features by their relevance to the output, thereby solving the problem of feature selection by constructing a learned continuum of feature inclusion-to-exclusion. The optimal compression of each feature—at every stage of approximation—allows fine-grained inspection of the distinctions among feature values that are most impactful for prediction. We develop a framework for extracting insight from the spectrum of approximate models and demonstrate its utility on a range of tabular datasets.

## 1 INTRODUCTION

Interpretability is a pressing issue for machine learning (ML) (Doshi-Velez & Kim, 2017; Fan et al., 2021; Rudin et al., 2022). As models continue to grow in complexity, machine learning is increasingly integrated into fields where flawed decisions can have serious ramifications (Caruana et al., 2015; Rudin et al., 2018; Rudin, 2019). Interpretability is not a binary property that machine learning methods have or do not: rather, it is the degree to which a learning system can be probed and comprehended (Doshi-Velez & Kim, 2017). Importantly, interpretability can be attained along many distinct routes (Lipton, 2018). Various constraints on the learning system can be incorporated, such as to force feature effects to combine in a simple (e.g., linear) manner, restricting the space of possible models in exchange for a degree of comprehensibility (Molnar et al., 2020; Molnar, 2022). In contrast to explainable AI that happens post-hoc after a black-box model is trained, interpretable methods engineer the constraints into the model from the outset (Rudin et al., 2022).

In this work, we introduce a novel route to interpretability that places no restrictions on model complexity, and instead tracks *how much* and *what* information is most important for prediction. By identifying optimal information from features of the input, the method grants a measure of salience to each feature, produces a spectrum of models utilizing different amounts of optimal information about the input, and provides a learned compression scheme for each feature that highlights from where the information is coming in fine-grained detail. The central object of interpretation is not a single

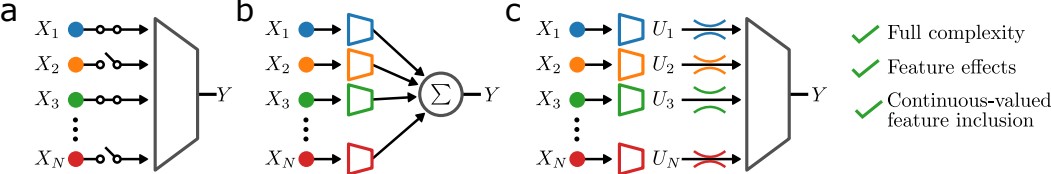

Figure 1: **Restricting information flow from features grants interpretability with full complexity.** **(a)** Feature selection methods optimize a binary mask on features to uncover the most informative ones. **(b)** Interpretable ML methods sacrifice model complexity in exchange for comprehensible feature interactions (shown is a generalized additive model (GAM) with a neural network to transform each feature). **(c)** Our method places an information bottleneck penalty on each feature, with no constraints on complexity before or after the bottlenecks. The features $X_i$ are compressed to optimal representations $U_i$, which communicate the most relevant information to the rest of the model. Compressing the features in this way allows feature inclusion/exclusion to exist on a continuum. The learned compression scheme for each feature grants insight into feature effects on the model output.

optimized model, but rather a family of models that reveals how predictive information is distributed across features at every level of fidelity.

The information constraint that we incorporate is a variant of the Information Bottleneck (IB) (Tishby et al., 2000; Asoodeh & Calmon, 2020). The IB extracts relevant information in a relationship, though it generally lacks interpretability because the extracted information is free to be any black-box function of the input. Here, we use the Distributed IB (Estella Aguerri & Zaidi, 2018; Murphy & Bassett, 2022), which siloes the information from each feature and applies a sum-rate constraint at an appropriate level in the model's processing: after every feature is processed and before any interaction effects between the features can be included. Before and after the bottlenecks, arbitrarily complex processing can occur, thereby allowing the Distributed IB to be used in scenarios with many features and complex feature interactions.

In this paper, we develop a framework for extracting insight from optimization with the Distributed IB. Through experiments on a variety of tabular datasets and comparisons to other approaches in interpretable ML and feature selection, we demonstrate the following strengths of the approach:

1. **Capacity to find the features *and* feature values that are most important for prediction.** The Distributed IB optimally allocates information across the features for every degree of approximation. Each feature is compressed during optimization, and the compression schemes reveal the distinctions among feature values that are most important for prediction.

2. **Full complexity of feature processing and interactions.** Compatible with arbitrary neural network architectures before and after the bottlenecks, the method gains interpretability without sacrificing predictive accuracy.

3. **Intuitive to implement, minimal additional overhead.** The method is a probabilistic encoder for each feature, employing the same loss penalty as $\beta$-VAE (Higgins et al., 2017). A single model is trained once while annealing the magnitude of the information bottleneck.

## 2 RELATED WORK

We propose a method that analyzes the information content of the features of an input with respect to an output. The method's generality bears relevance to several broad areas of research.

**Feature selection and variable importance.** Feature selection methods isolate the most important features in a relationship (Fig. 1a), both to increase interpretability and to reduce computational burden (Chandrashekar & Sahin, 2014; Cai et al., 2018). When higher order feature interactions are present, the large combinatorial search space presents an NP-hard optimization problem (Chandrashekar & Sahin, 2014). Modern feature selection methods navigate the search space with feature inclusion weights, or masks, that can be trained as part of the full stack with the machine learning model (Fong & Vedaldi, 2017; Balın et al., 2019; Lemhadri et al., 2021; Kolek et al., 2022).

Many classic methods of statistical analysis attempt to identify the most important variation in data. Canonical correlation analysis (CCA) (Hotelling, 1936) finds linear transformations of random variables that maximize linear correlations in the transformed space. The class of transformations can be extended to include kernel methods (Hardoon et al., 2004) and deep learning (Andrew et al., 2013; Wang et al., 2016). The desired linear correlation can be generalized to mutual information instead (Painsky et al., 2018). Relatedly, analysis of variance (ANOVA) finds a decomposition of variance by features (Fisher, 1936; Stone, 1994), and recent extensions incorporate ML-based transforms (Märtens & Yau, 2020). A similar set of methods known as variable importance (VI) quantify the effect of different features on an outcome (Breiman, 2001; Fisher et al., 2019).

**Interpretable machine learning.** Interpretability is often categorized along various binaries. One of those binaries is as either *intrinsic* or *post hoc* (Molnar et al., 2020). Intrinsic interpretability occurs when the route to comprehensibility is engineered into the learning model from the outset, as in this work. By contrast, post-hoc interpretability is constructed around an already-trained model (Rudin et al., 2022). Another distinction is *local* versus *global*. Local interpretability is comprehensibility built up piecemeal by inspecting individual examples (e.g., LIME (Ribeiro et al., 2016)), whereas for global interpretability comprehensibility applies to the whole space of inputs at once, as in this work.

Intrinsic interpretabilty is generally achieved through constraints that have been incorporated to reduce the complexity of the system. For example, generalized additive models (GAMs) require feature effects to combine linearly (Fig. 1b), though each effect can be an arbitrary function of a feature (Chang et al., 2021; Molnar, 2022). By using neural networks to learn the transformations from raw feature values to effects, neural additive models (NAMs) maintain the interpretability of GAMs with more expressive effects (Agarwal et al., 2021). Pairwise interactions between effects can be included for more complex models (Yang et al., 2021; Chang et al., 2022), though complications arise from degeneracy and the growing number of effect terms to be inspected. Regularization can be used to encourage sparsity in the ML model so that fewer components—weights in the network (Ng, 2004) or even features (Lemhadri et al., 2021)—conspire to produce the final prediction.

**Information Bottleneck.** Our proposed method can be seen as a regularization to encourage sparsity of information. It is based upon the Information Bottleneck, a method to extract the most relevant information in a relationship (Tishby et al., 2000) that has proven useful in representation learning (Alemi et al., 2016; Wang et al., 2019; Asoodeh & Calmon, 2020). As a form of post-hoc interpretability, Bang et al. (2021) and Schulz et al. (2020) employ the IB to explain a trained model's predictions. The IB's potential for intrinsic interpretability is hindered by the fact that it has only one bottleneck for the entire input, obscuring from where relevant information originated. By contrast, our method based on multiple IBs affects the training of a model itself to provide intrinsic interpretability.

Distributing IBs to multiple sources of information has been of interest in information theory as a solution to multiterminal source coding, or the "CEO problem" (Berger et al., 1996; Estella Aguerri & Zaidi, 2018; Aguerri & Zaidi, 2021; Steiner & Kuehn, 2021). The problem is the optimal compression of multiple signals independently before transmission to a "CEO" for prediction of a target variable. In Murphy & Bassett (2022), features of data were used as the distributed sources of information so that the Distributed IB revealed the relative importance of the features. Here we expand the framework of interpretability by analyzing the learned compression schemes for insight to feature value importance, and by benchmarking with related interpretability and feature selection methods.

## 3 METHODS

Conceptually, the Distributed IB introduces a penalty on the amount of information used by a black-box model about each feature of the input. Independently of all others, features are compressed and communicated through a learned noisy channel; the messages are then combined for input into a prediction network. The information allocation and compression schemes—for every feature and every magnitude of the penalty—serve as windows into feature importance and into feature effects.

### 3.1 DISTRIBUTED VARIATIONAL INFORMATION BOTTLENECK

Let $X, Y \sim p(x, y)$ be random variables representing the input and output of a relationship to be modeled from data. Mutual information is a measure of the statistical dependence between two random variables; it is the reduction of entropy of one variable after learning the value of the other:

$$I(X; Y) = H(Y) - H(Y|X), \tag{1}$$

where $H(Z) = \mathbb{E}_{z \sim p(z)}[-\log p(z)]$ is Shannon's entropy (Shannon, 1948).

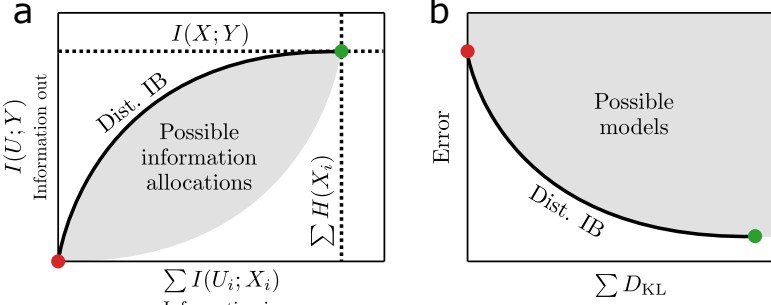

Figure 2: **The tradeoff between information into a model and predictive power out.** **(a)** The distributed information plane displays the sum total information from all features, encoded independently of one another, against the information about the output $Y$. While there are many possible allocations of feature information (gray region), the Distributed IB objective allows us to find a continuum of allocations (black curve) that contain the most information about the output for the least information about the input. Increasing the bottleneck strength $\beta$ traverses the optimal trajectory from the green point—all bottlenecks fully open, and all information is available for prediction—to the red point—where predictions are independent of the input. **(b)** A pragmatic alternative to **(a)** displays the sum of the KL-divergences $\sum D_{\mathrm{KL}}$ (roughly corresponding to the information utilized by the model about the features) against the particular error metric being optimized. Optimizing the Distributed IB objective finds a continuum of models (black curve) that utilize the least information about the features for the least error. In the large $\sum D_{\mathrm{KL}}$ regime, the effect of the bottlenecks are negligible and the models are essentially unconstrained machine learning. A Distributed IB optimization gradually increases $\beta$, the strength of the bottlenecks, shedding information.

The Information Bottleneck (Tishby et al., 2000) is a learning objective to extract the most relevant information from $X$ about $Y$. It optimizes a representation $U$ that conveys information about $X$ in the form of a stochastic channel of communication $U \sim p(u|x)$. The representation is found by minimizing a loss with competing mutual information terms balanced by a scalar parameter $\beta$,

$$\mathcal{L}_{\mathrm{IB}} = \beta I(U; X) - I(U; Y). \tag{2}$$

The first term is the bottleneck, a penalty on information passing into $U$, whose strength is controlled by the parameter $\beta$. Optimization with different values of $\beta$ produces different models of the relationship between $X$ and $Y$; a single sweep over $\beta$ produces a continuum of models that utilize different amounts of information about the input $X$. In the limit where $\beta \to 0$, the bottleneck is fully open and all information from $X$ may be freely conveyed into $U$ in order for it to be maximally informative about $Y$. As $\beta$ increases, only the most relevant information in $X$ about $Y$ becomes worth conveying into $U$, until eventually an optimal $U$ is uninformative random noise.

One challenge in using the IB is that mutual information is generally difficult to estimate from data (Saxe et al., 2019; McAllester & Stratos, 2020). Accordingly, variants of IB replace the mutual information terms of Eqn. 2 with bounds amenable to deep learning (Alemi et al., 2016; Achille & Soatto, 2018; Kolchinsky et al., 2019). Here we follow the variational information bottleneck (VIB) (Alemi et al., 2016), whose framework resembles that of $\beta$-VAEs (Higgins et al., 2017).

A sample $x$ is encoded as a distribution in representation space $p(u|x) = f(x, \phi, \epsilon)$ with a neural network parameterized by weights $\phi$ and a source of noise $\epsilon \sim \mathcal{N}(0, 1)$ to facilitate the "reparameterization trick" (Kingma & Welling, 2014). The bottleneck restricting the information from $X$ takes the form of the Kullback-Leibler (KL) divergence between the encoded distribution $p(u|x)$ and a prior distribution $r(u) = \mathcal{N}(0, 1)$. As the KL divergence tends to zero, all representations are indistinguishable from the prior and from each other, and therefore uninformative. A representation $u$ is sampled from $p(u|x)$ and then decoded to a distribution over the output $Y$ by a second neural network parameterized by weights $\psi$, $q(y|u) = g(u, \psi)$. In place of Eqn. 2, the following may be minimized with standard gradient descent methods:

$$\mathcal{L}_{\mathrm{VIB}} = \beta D_{\mathrm{KL}}(p(u|x)\|r(u)) - \mathbb{E}[\log q(y|u)]. \tag{3}$$

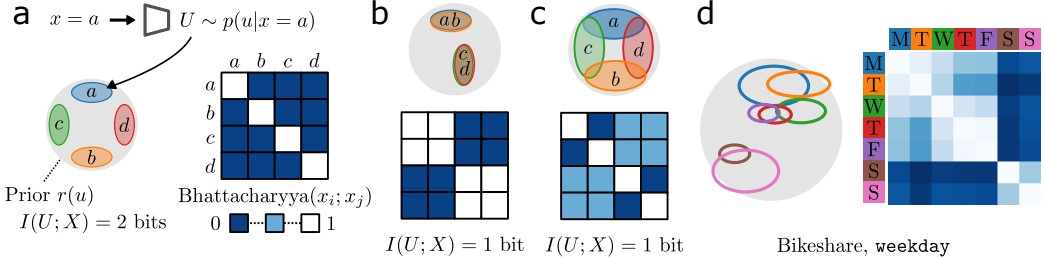

Figure 3: **Confusion matrices to visualize compression schemes**. **(a)** An input value is encoded to a distribution over $u$. The Bhattacharyya coefficient quantifies how distinguishable the distributions are for different encodings, which we express as a matrix. The diagonal (white) values are indistinguishable and everything else (blue) is distinguishable, yielding 2 bits of information. **(b)** 1 bit of information allows a hard clustering of the values, such that the $ab$ cluster is distinguishable from the $cd$ cluster, but no more than that. **(c)** 1 bit also allows a soft clustering, where some pairs of values are partially distinguishable. **(d)** When trained with a two-dimensional embedding space per feature on the Bikeshare dataset, we recover a soft clustering of the day of the week. Weekend is distinguishable from weekday, with intermediate values of distinguishability otherwise.

The Distributed IB (Estella Aguerri & Zaidi, 2018) concerns the optimal scheme of compressing multiple sources of information as they relate to a relevance variable $Y$. Let $\{X_i\}$ be the features of $X$; we use the Distributed IB to encode each feature $X_i$ independently of the rest, and then integrate the information from all features to predict $Y$. A bottleneck is installed after each feature in the form of a compressed representation $U_i$ (Fig. 1c), and the set of representations $U_X = \{U_i\}$ is used as input to a model to predict $Y$. The same mutual information bounds of the Variational IB (Alemi et al., 2016) can be applied in the Distributed IB setting (Aguerri & Zaidi, 2021),

$$\mathcal{L}_{\text{DIB}} = \beta \sum_i D_{\text{KL}}(p(u_i|x_i)||r(u_i)) - \mathbb{E}[\log q(y|u_X)]. \tag{4}$$

**Arbitrary error metrics with feature information regularization.** Often it may be more natural to quantify predictive power with an error metric other than cross entropy; the information penalty on each feature remains unchanged. For instance, mean squared error (MSE) may be used instead,

$$\mathcal{L} = \beta \sum D_{\text{KL}} + ||y - \hat{y}||^2, \tag{5}$$

where $y$ and $\hat{y}$ are the prediction and ground truth values, respectively. $\beta$ is no longer a unitless ratio of information about $Y$ to the sum total information about the features of $X$, and the optimal range must be found anew for each problem. Optimization will nevertheless yield a spectrum of approximations that prioritizes the most relevant information about the input features.

**Boosting expressivity for low-dimensional features.** Consistent with results reported by Agarwal et al. (2021), we found that the feature encoders struggled with expressivity due to the low dimensionality of the features. We *positionally encode* feature values (Tancik et al., 2020) before they pass to the neural network, taking each value $z$ to $[\sin(\omega_1 z), \sin(\omega_2 z), ...]$ with $\omega_k$ the encoding frequencies.

### 3.2 VISUALIZING THE OPTIMALLY PREDICTIVE INFORMATION

**Distributed information plane.** Tishby & Zaslavsky (2015) popularized the visualization of learning dynamics in the so-called information plane, a means to display how successfully information conveyed about the input of a relationship is converted to information about the output. Here we employ the distributed information plane (Fig. 2a) to compare the total information utilized about the features $\sum I(U_i; X_i)$ to information about the output $I(\{U_i\}; Y)$. In the space of possible allocations of information to the input features, the Distributed IB finds those that minimize $\sum I(U_i; X_i)$ while maximizing $I(\{U_i\}; Y)$ (black curve, Fig. 2a).

A pragmatic alternative to the distributed information plane replaces $\sum I(U_i; X_i)$ with its upper bound used during optimization: the sum of KL divergences across the features. Additionally, to be compatible with arbitrary error metrics, we replace the vertical axis $I(\{U_i\}; Y)$ with the error used for training.

Table 1: Performance on various tabular datasets. NAM and NODE-GAM allow no feature interactions; NODE-GA$^2$M contain pairwise feature interactions; the Neural Oblivious Decision Ensemble (NODE) LassoNet, and Distributed IB methods are full complexity models (allowing arbitrary feature interactions). ‡Results from Chang et al. (2022).

| | Bikeshare (RMSE ↓) | Wine (RMSE ↓) | MIMIC-II (AUC % ↑) | MIMIC-III (AUC % ↑) | Microsoft (MSE ↓) |
|---|---|---|---|---|---|
| NAM (Agarwal et al., 2021) | $99.9_{\pm 1.3}$ | $0.71_{\pm 0.02}$ | $83.0_{\pm 0.8}$ | $81.7_{\pm 0.2}$ | $0.5908_{\pm 0.0004}$ |
| NODE-GAM (Chang et al., 2022) | $100.7_{\pm 1.6}$ | $0.71_{\pm 0.03}$ | $83.2_{\pm 1.1}$ | $81.4_{\pm 0.5}$ | $0.5821_{\pm 0.0004}$ |
| NODE-GA$^2$M (Chang et al., 2022) | $49.8_{\pm 0.8}$ | $0.67_{\pm 0.02}$ | $84.6_{\pm 1.1}$ | $82.2_{\pm 0.7}$ | $0.5618_{\pm 0.0003}$ |
| LassoNet (Lemhadri et al., 2021) | $46.0_{\pm 2.0}$ | $0.68_{\pm 0.02}$ | $83.4_{\pm 0.8}$ | $79.4_{\pm 0.8}$ | $0.5667_{\pm 0.0012}$ |
| NODE‡ (Popov et al., 2019) | $\mathbf{36.2}_{\pm 1.9}$ | $0.64_{\pm 0.01}$ | $84.3_{\pm 1.1}$ | $\mathbf{82.8}_{\pm 0.7}$ | $\mathbf{0.5570}_{\pm 0.0002}$ |
| Distributed IB (Ours) | $38.0_{\pm 1.1}$ | $\mathbf{0.62}_{\pm 0.01}$ | $\mathbf{84.9}_{\pm 1.2}$ | $80.2_{\pm 0.4}$ | $0.5790_{\pm 0.0003}$ |

The entire curve of optimal information allocations in Fig. 2b may be found in a single training run, by increasing $\beta$ logarithmically over the course of training as Eqn. 4 is optimized. Training begins with negligible information restrictions ($\beta \sim 0$) so that the relationship may be found without obstruction (Wu et al., 2020; Lemhadri et al., 2021), and ends once all information is removed. Each value of $\beta$ corresponds to a different approximation to the relationship between $X$ and $Y$, and we can use the KL divergence for each feature as a measure of its overall importance in the approximation.

**Confusion matrices to visualize feature compression schemes.** Transmitting information about a feature is equivalent to conveying distinctions between the feature's possible values. In the course of optimizing the Distributed IB, the spectrum of learned models range from all input values perfectly distinguishable ($\beta \to 0$, unconstrained machine learning) to all inputs perfectly indistinguishable (the model utilizes zero information about the input). Along the way, the distinctions most important for predicting $Y$ are selected, and can be inspected via the encoding of each feature value.

Each feature value $x_i$ is encoded to a probability distribution $p(u_i|x_i)$ in channel space, allowing the compression scheme to be visualized with a confusion matrix showing the similarity of feature values (Fig. 3a). The Bhattacharyya coefficient (Kailath, 1967) is a symmetric measure of the similarity between probability distributions $p(z)$ and $q(z)$, and is straightforward to compute for normal distributions. The coefficient is 0 when the distributions are perfectly disjoint (i.e., zero probability is assigned by $p(z)$ everywhere that $q(z)$ has nonzero probability, and vice versa), and 1 when the distributions are identical. In other words, if the Bhattacharyya coefficient between encoded distributions $p(u|x = a)$ and $p(u|x = b)$ is 1, the feature values $x = a$ and $x = b$ are indistinguishable to the prediction network and the output is independent of whether the actual feature value was $a$ or $b$. On the other extreme where the Bhattacharyya coefficient is 0, the feature values are perfectly distinguishable and the prediction network can yield distinct outputs for $x = a$ and $x = b$.

We display in Fig. 3b,c possible compression schemes containing one bit of information about a hypothetical $x$ with four equally probable values. The compression scheme can be a hard clustering of the values into two sets where $x = a$ and $x = b$ are indistinguishable, and similarly for $x = c$ and $x = d$ (Fig. 3b). Another scheme with one bit of information is the soft clustering shown in Fig. 3c, where each feature value is partially 'confused' with other feature values. In the case of soft clustering, a learned compression cannot be represented by a decision tree; rather, the confusion matrices are the fundamental distillations of the information preserved about each feature. An example in Fig. 3d shows the learned compression scheme of the `weekday` feature in the Bikeshare dataset (details below). The soft clustering distinguishes between the days of the week to different degrees, with the primary distinction between weekend days and weekdays.

## 4 EXPERIMENTS

We analyzed a variety of tabular datasets with the Distributed IB and compared the interpretability signal to what is obtainable through other methods. Our emphasis was on the unique capabilities of the Distributed IB to produce insight about the relationship in the data modelled by the neural network. Performance results are included in Tab. 1 for completeness. Note that the Distributed IB places no constraints on the machine learning models used and in the $\beta \to 0$ regime could hypothetically be as good as any finely tuned black-box model. The benefit of freedom from complexity constraints

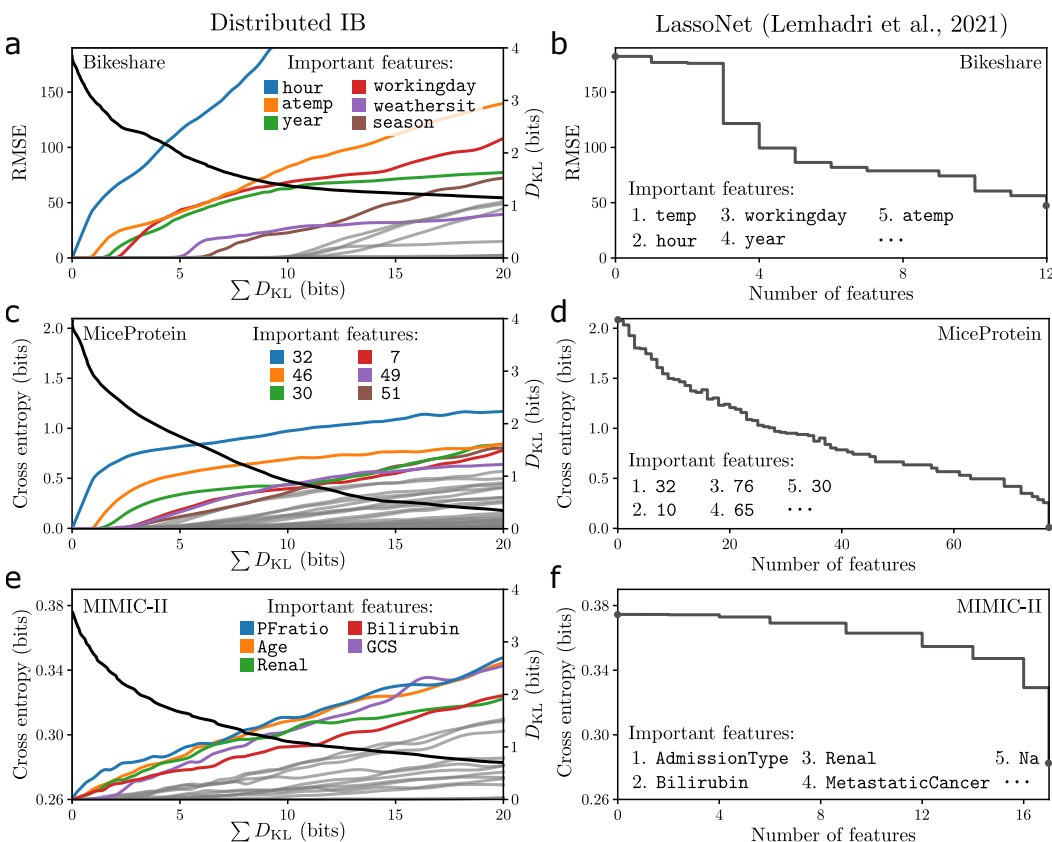

Figure 4: **Distributed information plane trajectories with information allocation by feature. (a)** For the Bikeshare dataset, the optimal RMSE error versus $\sum D_{\mathrm{KL}}$ (black) decays as more information is used by the model about the features. The shifting allocation of information to the features tells of their relevance to the prediction at each level of approximation. Only a few bits about a handful of features is enough for most of the performance. We label and color the first features to contribute non-negligibly to the approximations. **(b)** The LassoNet trajectory for the Bikeshare dataset yields only a discrete series of error values and an ordering of the features. Note the LassoNet trajectory extends to the point of all features included, whereas the Distributed IB is truncated at 20 bits of $\sum D_{\mathrm{KL}}$. **(c, d)** Same as **(a, b)** for the MiceProtein dataset, with expression levels of 77 proteins measured to classify mice in one of eight categories related to Down syndrome. **(e, f)** Same as **(a, b)** for the MIMIC-II dataset, a binary classification task related to ICU mortality, with 17 input features.

was particularly clear in the smaller regression datasets, whereas performance was only middling on several other datasets (refer to App. B for extended results and App. C for implementation details).

## 4.1 THE FEATURES WITH THE MOST PREDICTIVE INFORMATION

We used the Distributed IB to identify informative features for three datasets in Fig. 4. Bikeshare (Dua & Graff, 2017) is a regression dataset to predict the number of bikes rented given 12 descriptors of the date, time, and weather. MiceProtein (Higuera et al., 2015) is a classification dataset relating the expression of 77 genes to normal and trisomic mice exposed to various experimental conditions. MIMIC-II (Johnson et al., 2016) is a binary classification dataset to predict hospital ICU mortality given 17 patient measurements. We compared our results using the Distributed IB to those obtained using LassoNet (Lemhadri et al., 2021), a feature selection method that attaches an L1 regularization penalty to each feature and introduces a proximal algorithm into the training loop to enforce a constraint on weights. By gradually increasing the regularization, LassoNet finds a series of models that utilize fewer features while maximizing predictive accuracy, providing a signal about what features are necessary for different levels of accuracy.

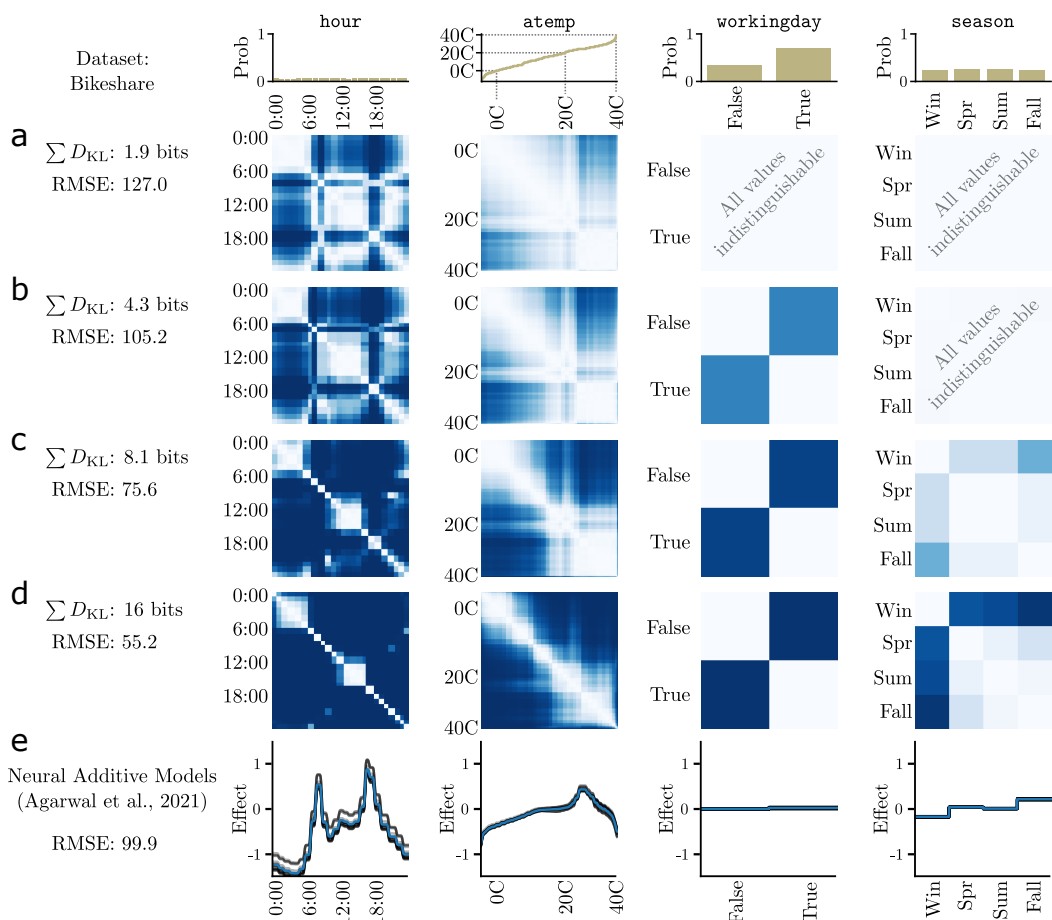

Figure 5: **Important information within each feature via inspection of the learned compression schemes. (a-d)** For the Bikeshare dataset at different levels of approximation, confusion matrices display the learned compression schemes for four of the top features found in Fig. 4. Following the scheme in Fig. 3, the pairwise distinguishability of feature values visualizes the information preserved about each feature. For the categorical features (hour, workingday, and season), each row and column of the confusion matrices corresponds to a unique value, and the probability distributions at the top of the figure are aligned with the columns of the matrices. For the continuous variable atemp (apparent temperature), 1000 values were sampled randomly from the dataset and sorted for use as the inspected points; the rank order plot at the top shows their order in the matrices. As the utilized information $\sum D_{\mathrm{KL}}$ increases, so too does the performance; with only four bits the Distributed IB can perform about as well as NAM, highlighting the value of using a full-complexity model. **(e)** The effects of these features on the output can be directly visualized with NAM; the horizontal coordinates match those of the matrices in **(a-d)** and the black traces display effects for an ensemble of 25 NAMs.

Like LassoNet, the Distributed IB probes the tradeoff between information about the features and predictive performance, though with important differences. LassoNet can only vary the information used for prediction through a discrete-valued proxy: the number of features. Binary inclusion or exclusion of features is a common limitation for feature selection methods (Bang et al., 2021; Kolek et al., 2022). By contrast, the Distributed IB compresses each feature along a continuum, selecting the most informative information from across all input features. At every point along the continuum is an allocation of information to each of the features. For example, in Fig. 4a we found that only a few bits of information ($\sum D_{\mathrm{KL}}$), largely deriving from the hour feature, was necessary to achieve most of the performance of the full-information model, and to outperform the linear models (Tab. 1). The capacity to extract partial information from features is notable in comparison to the LassoNet performance (Fig. 4b), which improves only after multiple complete features have been incorporated.

In contrast to Bikeshare, where decent approximations required only a handful of features, the MiceProtein (Fig. 4c,d) and MIMIC-II (Fig. 4e,f) datasets were found to require information from many features simultaneously. The information allocation is dynamic, with some features saturating quickly, and others growing in information usage at different rates. From these observations, we see that the signal obtained by the Distributed IB can usefully decompose datasets with a large number of features and complex interaction effects, all within a single training run.

## 4.2    Fine-grained inspection of feature effects

In addition to a measure of feature importance in the form of information allocation, the Distributed IB offers a second source of interpretability: the relative importance of particular feature values, revealed through the learned compression schemes of each feature. One of the primary strengths of a GAM is that the effects of the feature values are straightforwardly obtained, though at the expense of interactions between the features. By contrast, the Distributed IB permits arbitrarily complex feature interactions and still maintains a window into the role played by specific feature values in the ultimate prediction. In Fig. 5 we compared the compression schemes found by the Distributed IB to the feature effects obtained with NAM (Agarwal et al., 2021) on the Bikeshare dataset.

The Distributed IB allows one to build up to the maximally predictive model by approximations that gradually incorporate information from the input features. To a first approximation (Fig. 5a), with only two bits of information about all features, the scant distinguishing power was allocated to discerning certain times of day—commuting periods, the middle of the night, and everything else—in the `hour` feature. Some information discerned hot from cold from the apparent temperature (`atemp`), and no information about the season or whether it was a working day was used for prediction. We are unable to say how this information was used for prediction—the price of allowing the model to be more complex—but we can infer much from where the discerning power was allocated. At four bits, and performing about as well as the linear models in Tab. 1, the Boolean `workingday` was incorporated for prediction, and the evening commuting time was distinguished from the morning commuting time. With eight and sixteen bits of information about the inputs, nearly all of the information was used from the `hour` feature, the apparent temperature became better resolved, and in `season`, winter was distinguished from the warmer seasons.

The feature effects found with NAM (Fig. 5e) partially corroborated the distinguishability found by the Distributed IB, particularly in the `hour` feature (e.g., commuting times, the middle of the night, and the rest form characteristic divisions of the day in terms of the NAM effect) . However, we found that the interpretability of the NAM-effects tend to vary significantly across datasets (see App. B for more examples). For example, the apparent temperature (`atemp`) and temperature (`temp`) features were found to have contradictory effects (Fig. 10) that were difficult to comprehend, presumably arising from feature interactions that NAM could not model. Thus while the Distributed IB sacrifices a direct window to feature effects, tracking the information on the input side of a black-box model may serve as one of the only sources of interpretability in cases with complex feature interactions.

## 5    Discussion

Interpretability can come in many forms (Lipton, 2018); the Distributed IB provides a novel source of interpretability by illuminating where information resides in a relationship with regards to features of the input. Without sacrificing complexity, the Distributed IB is particularly useful in scenarios with large numbers of features and complex interaction effects. The capacity for successive approximation, allowing detail to be incorporated gradually so that humans' limited cognitive capacity (Cowan, 2010; Rudin et al., 2022) does not prevent comprehension of the full relationship, plays a central role in the Distributed IB's interpretability.

The signal from the Distributed IB—particularly the compression schemes of feature values (e.g., Fig. 5)—is vulnerable to over-interpretation. This vulnerability is partly due to a difficulty surrounding intuition about compression schemes (i.e., it is perhaps most natural to imagine 1 bit of information as a single yes/no question, or a hard clustering (Fig. 3b), rather than as a confusion matrix (Fig. 3c)). It is also partly due to the inability to make direct claims about how a particular feature value changes the output. Rather than supplanting other interpretability methods, the Distributed IB should be seen as a new tool for interpretable machine learning, supplementing other methods. A thorough analysis could incorporate methods where the effects are comprehensible, possibly using a reduced feature set found by the Distributed IB.

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

## Appendices for *Interpretability with full complexity by constraining feature information*

## A SYNTHETIC EXAMPLE

We work through a synthetic example with the Distributed IB in Fig. 6. There are two binary-valued interventions $A, B$ which we take to be the features $X_1, X_2$ of the input $X$, affecting a binary outcome $Y$ with a joint probability distribution shown in Fig. 6a. Using 10,000 samples from the joint distribution, we train the Distributed IB and recover the distributed information plane (Fig. 6b) revealing that intervention A is more important for predicting the outcome $Y$.

Each feature is embedded to a two-dimensional Gaussian so that they may be directly visualized (Fig. 6c), alongside the computed confusion matrix. Note that the feature values of the interventions split along the cardinal axes because the embeddings are diagonal Gaussians. That intervention A split horizontally while B split vertically is due to chance.

## B EXTENDED VISUALIZATIONS

We show in Fig. 7 Distributed IB optimizations for the remaining three datasets mentioned in the manuscript, alongside their LassoNet counterparts. In Figs. 8 and 9 we show the compression schemes for some of the most important features in the Wine and MIMIC-II datasets. Finally, in Fig. 10, we display the compression schemes for all 12 features of the Bikeshare dataset over a dense sampling of information rates.

## C IMPLEMENTATION DETAILS

Code and additional examples may be found through the project page, distributed-information-bottleneck.github.io.

All Distributed IB experiments were run on a single computer with a 12 GB GeForce RTX 3060 GPU; 100k steps on the Bikeshare dataset takes a couple minutes, and on Microsoft about 2 hours.

The data loading and preprocessing is heavily based on the publicly released code from NODE-GAM (Chang et al., 2022).

We subclassed `tensorflow.keras.Model` so that training can leverage the `keras` high level API, and is only one step more difficult than the `model.compile(); model.fit()` flow. The additional step is the creation of a `keras Callback` to control the $\beta$ ramp over the course of training. All of the information allocations can be recovered from the `tensorflow.keras.callbacks.History` object returned at the end of training.

Training hyperparameters and architecture details are shown in Tab. 2 (common to all datasets) and Tab. 3 (dataset-specific). In the first stage of hyperparameter tuning, we explored parameter space in an unstructured manner and found that most parameters affected all datasets similarly. After some tuning, we settled on the values listed in Tab. 2. We found the effects of dropout rate and the number of annealing steps to differ by dataset, and selected optimal values out of $\{0, 0.1, 0.3, 0.5\}$ and $\{2.5 \times 10^4, 5 \times 10^4, 10^5\}$, respectively (Tab. 3).

### C.1 DATASETS

We used the dataset preprocessing code released with NODE-GAM (Chang et al., 2022) on Github[1] with a minor modification to use one-hot encoding[2] for categorical variables instead of leave-one-out encoding[3] when input to the Distributed IB models, unless there were more than 100 categories.

---

[1] https://github.com/zzzace2000/nodegam
[2] https://contrib.scikit-learn.org/category_encoders/onehot.html
[3] https://contrib.scikit-learn.org/category_encoders/leaveoneout.html

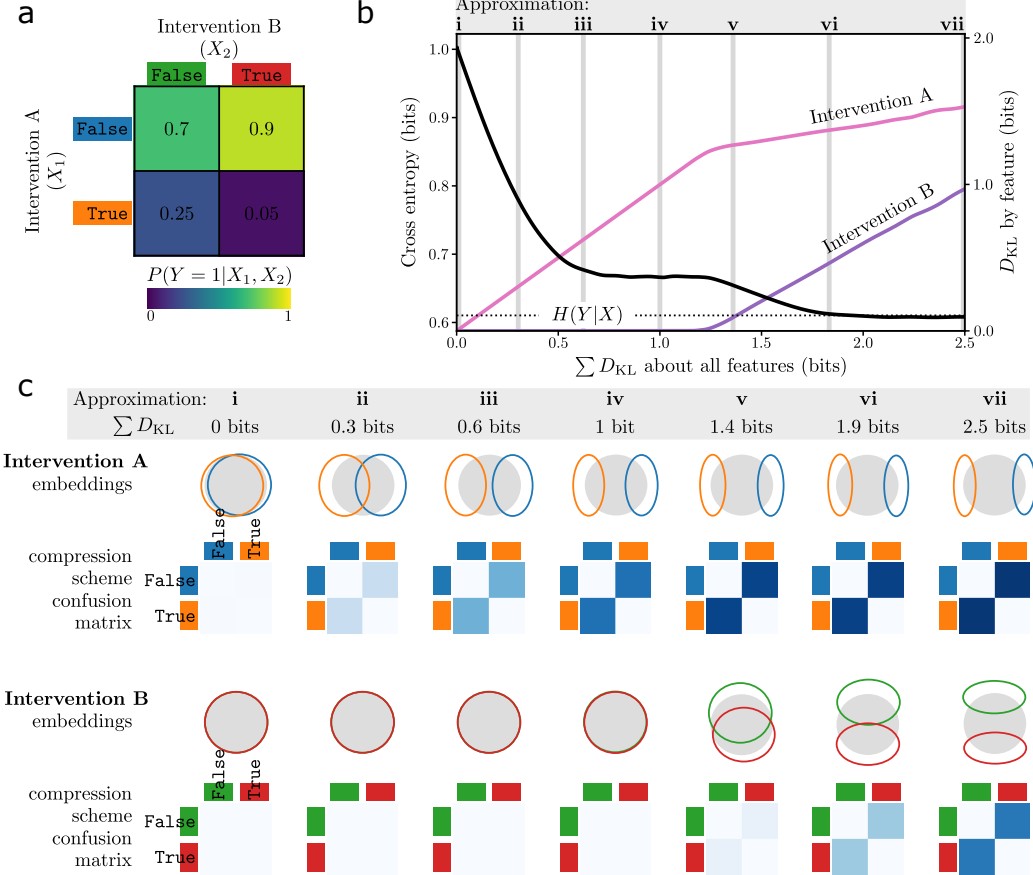

Figure 6: **Synthetic example: two interventions on a binary-valued outcome.** We visualize information allocation on a simple example. **(a)** Consider a hypothetical situation with two possible interventions, A and B, applied uniformly at random and affecting a binary outcome $Y$. The effects of applying the interventions can be visualized by a 2×2 table displaying $P(Y = 1|X)$. **(b)** By training the Distributed IB with 10,000 samples from the joint distribution represented by the table in **(a)**, we obtain a trajectory in the distributed information plane (black) showing the cross entropy error as a function of the amount of information incorporated about the interventions. Intervention A is found to be more important in predicting the outcome of $Y$. When information about intervention B is incorporated, the error eventually bottoms out at the conditional entropy $H(Y|X)$. **(c)** With each feature value embedded to a two-dimensional Gaussian distribution, we can visualize the embeddings alongside the computed confusion matrices. For the seven approximations highlighted in **(b)**, we display the learned compression schemes for interventions A (top) and B (bottom). As noted in the information allocation in **(b)**, the feature values of intervention A are distinguished first, and then the values of intervention B around approximation **v** and beyond.

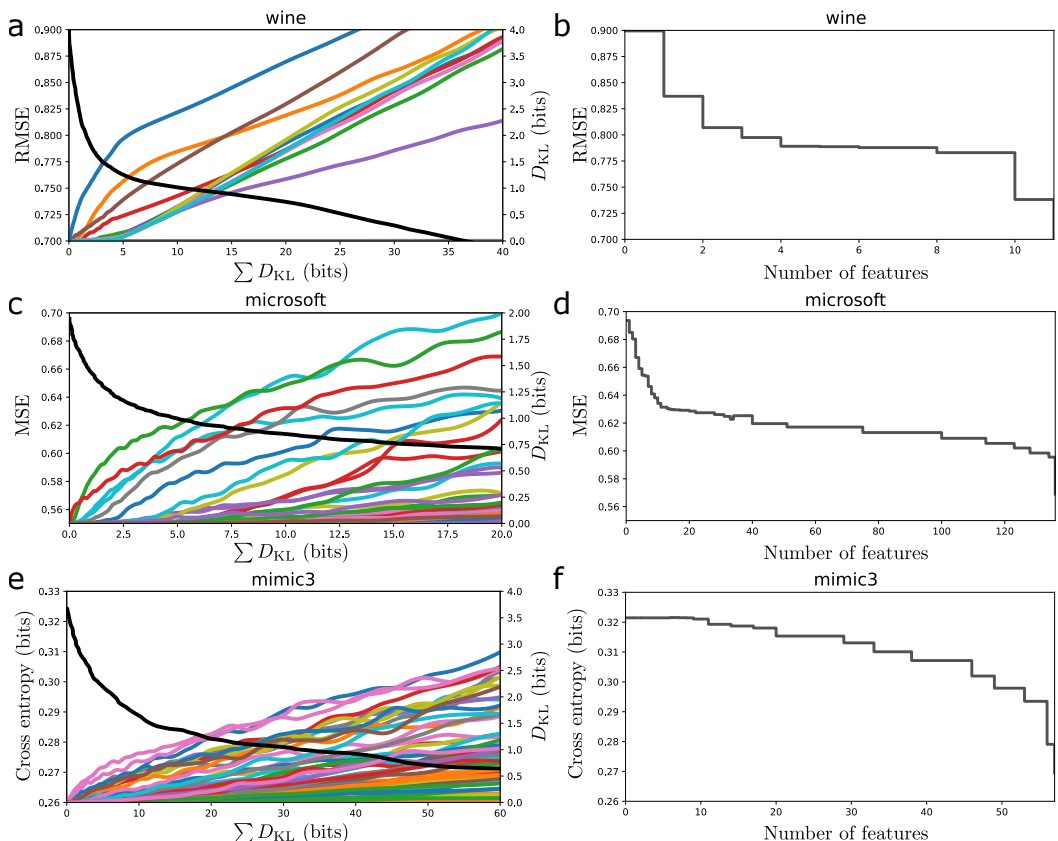

Figure 7: **Information plane optimizations for the remaining datasets.** We display Distributed IB optimizations alongside LassoNet results. Note the $\sum D_{\text{KL}}$ range is not uniform across datasets. **(a, b)** Wine, regress wine quality with 12 features (Dua & Graff, 2017); **(c, d)** Microsoft Learn to Rank, 136 features (Qin & Liu, 2013); **(e, f)** MIMIC-III, similar to MIMIC-II but with 57 features (Johnson et al., 2016).

| Parameter | Value |
|---|---|
| Positional encoding frequencies | [1, 2, 4, 8] |
| Nonlinear activation | Leaky ReLU ($\alpha = 0.2$) |
| Feature encoder MLP (one per feature) | [128, 128] |
| Bottleneck embedding space dimension | 8 |
| Joint encoder MLP | [256, 256] |
| Batch size | 128 |
| Optimizer | Adam |
| Learning rate | $3 \times 10^{-4}$ |
| $\beta_{\text{initial}}$ | $2 \times 10^{-5}$ |
| $\beta_{\text{final}}$ | 2 |

Table 2: Training parameters for the Distributed IB experiments.

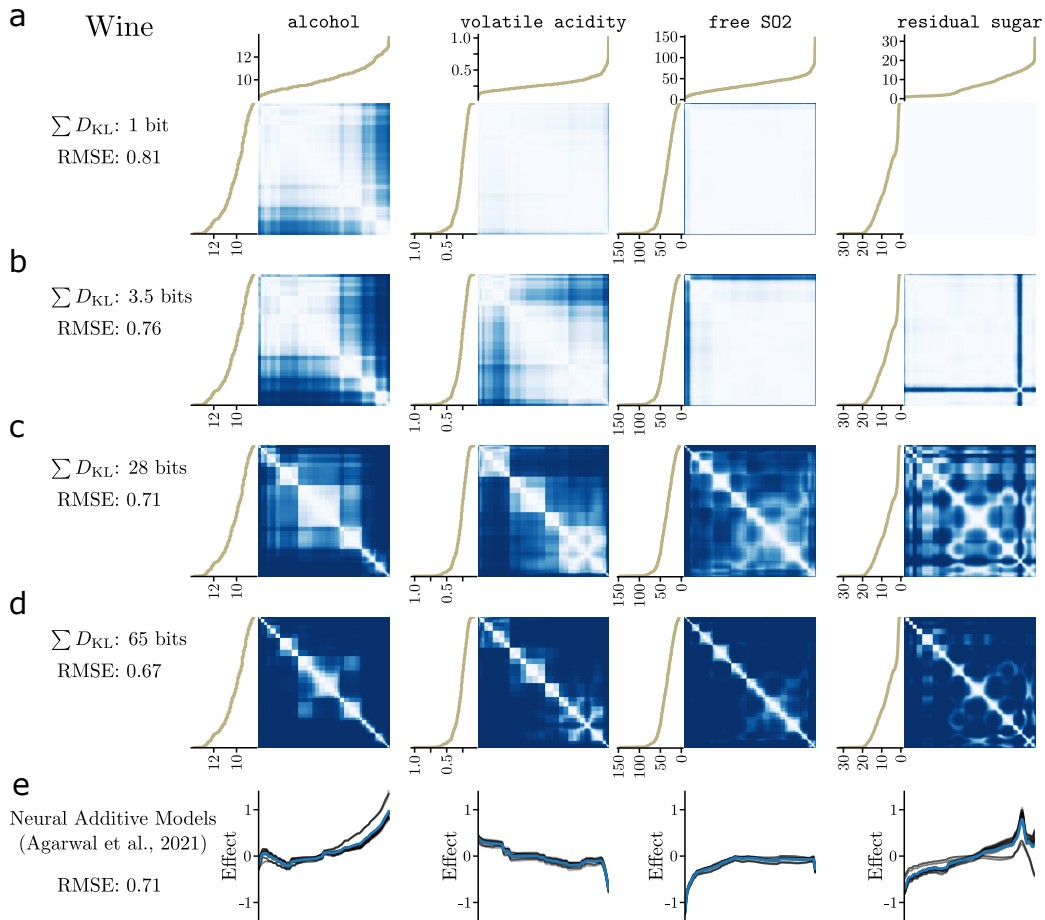

Figure 8: **Feature effects: Wine. (a-d)** For the Wine dataset, we display distinguishability matrices at different levels of approximation for four features. To the left (top) of the distinguishability matrices are the feature values corresponding to the row (column), which have been sorted. The features (`alcohol`, `volatile acidity`, `free SO2`, and `residual sugar`) were the first four to contribute to the approximations in the low $\sum D_{\mathrm{KL}}$ regime of Fig. 7a. As $\sum D_{\mathrm{KL}}$ increases, the RMSE improves. **(e)** The effects of these four features can be directly visualized with NAM; the horizontal coordinates match those of the matrices in **(a-d)** and the black traces display effects for an ensemble of 25 NAMs.

### C.2 COMPARISON METHODS

**LassoNet** (Lemhadri et al., 2021) We used the publicly released Pytorch code[4] with only minor modifications: (i) fixed a typo in `interfaces.py`[5] (delete '='), (ii) changed the objective outputs during training[6] to be solely the error metric (excluding the regularization terms) for the plots of Fig. 4.

We used the `LassoNetClassifier` and `LassoNetRegressor` classes, instead of the cross-validation versions that were released after the original publication, and performed our own cross-validation with the same splits for the other methods. For each dataset we tuned the following hyperparameters with a grid search: L2-regularization weight $\gamma \in \{0, 0.01, 0.1, 1\}$ and the dropout

---

[4] https://github.com/lasso-net/lassonet
[5] https://github.com/lasso-net/lassonet/blob/54d05048eca51c7b57077f0e09d4a72767f3e567/lassonet/interfaces.py#L470
[6] https://github.com/lasso-net/lassonet/blob/54d05048eca51c7b57077f0e09d4a72767f3e567/lassonet/interfaces.py#L251

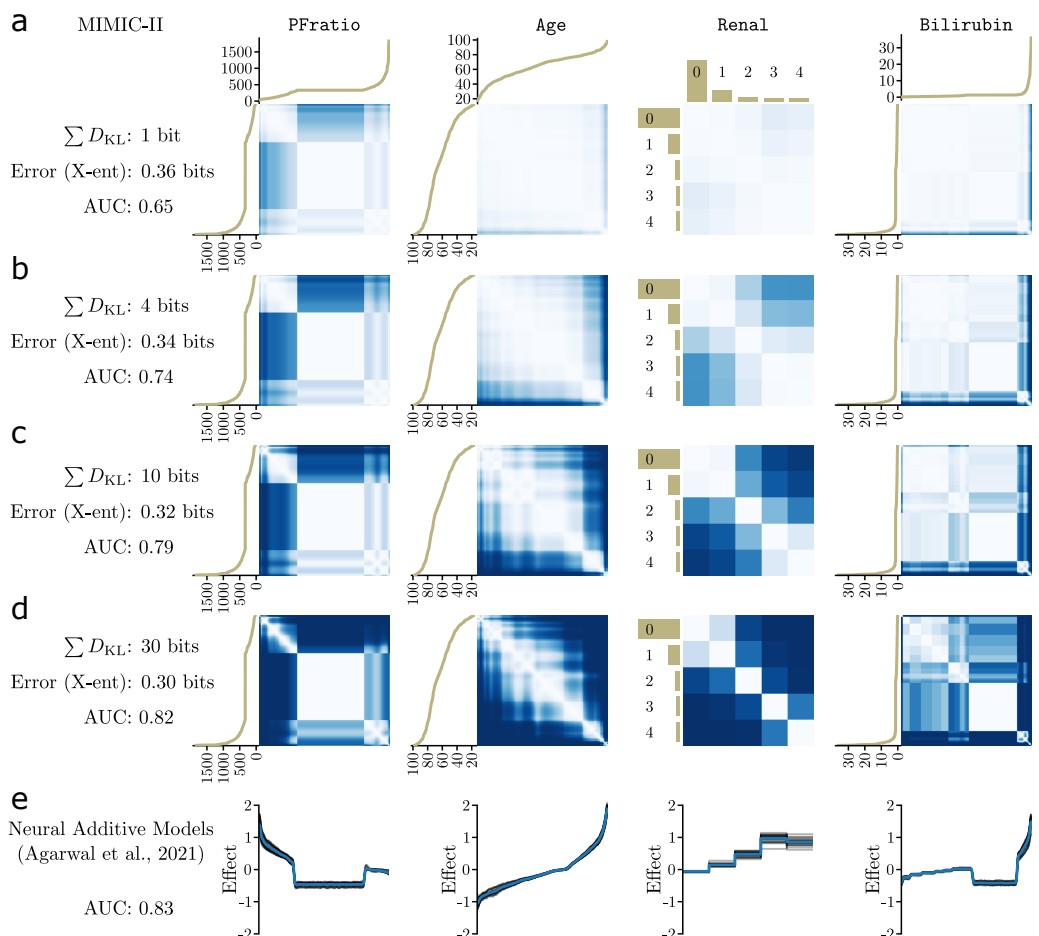

Figure 9: **Feature effects: MIMIC-II. (a-d)** For the MIMIC-II dataset, we display distinguishability matrices at different levels of approximation for four features. To the left (top) of the distinguishability matrices are the feature values corresponding to the row (column), which have been sorted. The features (PFratio, Age, Renal, and Bilirubin) were the first four to contribute to the approximations in the low $\sum D_{KL}$ regime of Fig. 4. For the categorical variable Renal the barchart displays the frequency of the particular values. As $\sum D_{KL}$ increases, the cross entropy error decreases and the ROC AUC (area under curve) increases. **(e)** The effects of these four features can be directly visualized with NAM; the horizontal coordinates match those of the matrices in **(a-d)** and the black traces display effects for an ensemble of 25 NAMs.

fraction in $\{0, 0.1, 0.3, 0.5\}$. We selected the parameters with the best integrated performance on the validation set (metric $\times$ number of features), and used the same hidden dimensions as the joint encoder for the Distributed IB method (2 layers of 256 units).

**Neural additive models**    (Agarwal et al., 2021) We used the publicly released Tensorflow code[7]. We modified the size of the feature encoders: we encountered memory issues with the default feature encoder sizes of $[M, 64, 32]$, with $M$ a function of the number of unique values in the training set for that feature[8]. Instead we made each feature encoder size [256, 256, 256] for the datasets with under 20 features, and [128, 128, 128] for the rest.

---

[7]https://github.com/google-research/google-research/tree/master/neural_additive_models

[8]https://github.com/google-research/google-research/blob/e79b8f09344935a2ab22660293233eb032df13a1/neural_additive_models/graph_builder.py#L283

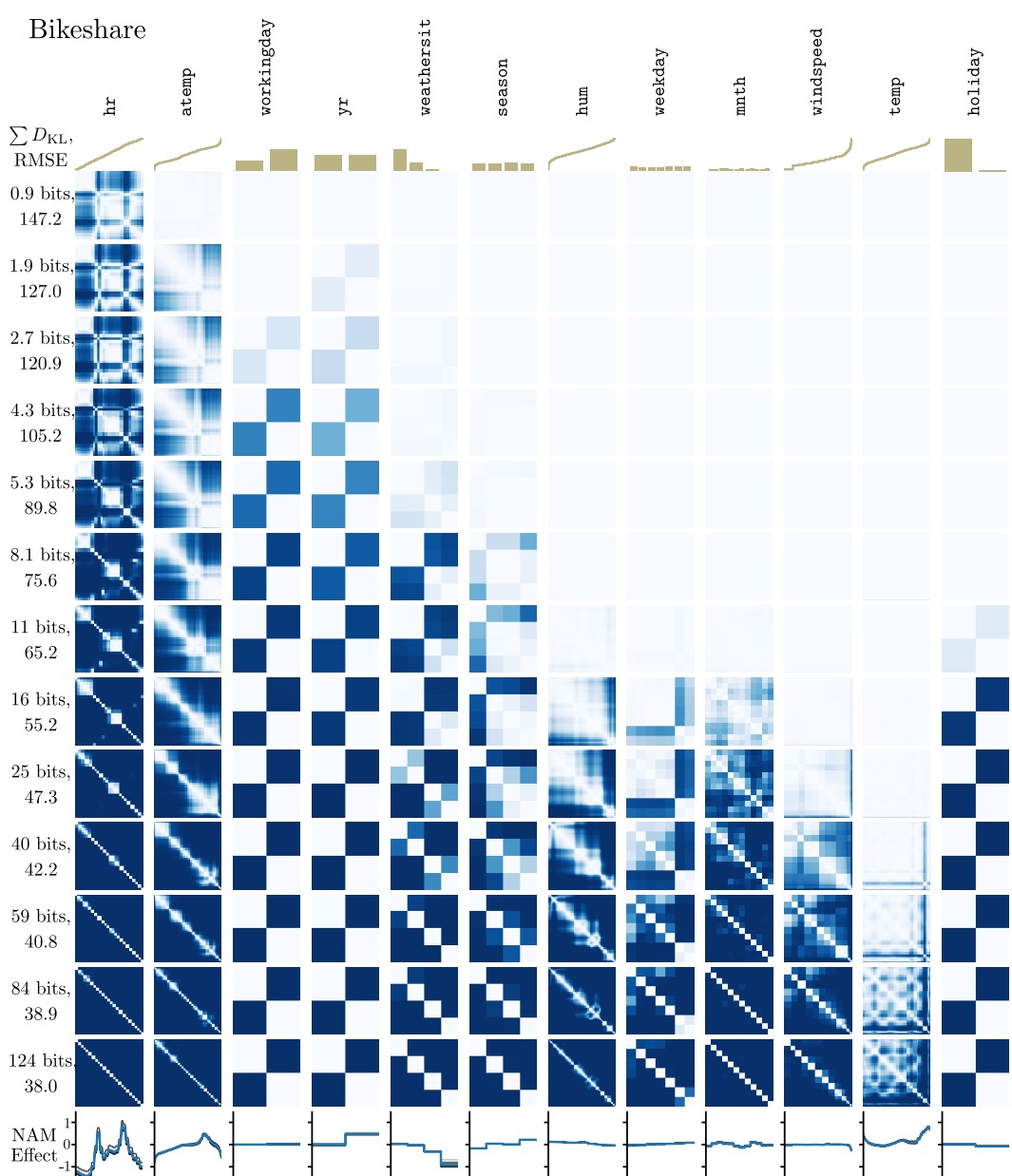

Figure 10: **Feature compression schemes: Bikeshare, extended.** We show the compression schemes for all 12 features of the Bikeshare dataset, for many values of $\sum D_{\mathrm{KL}}$, and compare in the bottom row to the feature effects found by NAM. The features were ordered left to right by the order of crossing a threshold value in Figure 4. Note the gradual addition of features and of distinguishability as the information rate increases. Eventually most of the feature value confusion matrices tend toward diagonal matrices, signifying that each unique feature value is distinguishable from all others in the compression scheme.

For each dataset we tuned the following hyperparameters with a grid search: hidden activation in $\{\texttt{relu}, \texttt{exu}\}$, dropout in $\{0, 0.1, 0.3\}$, weight decay (L2 regularization) in $\{0, 10^{-5}, 10^{-4}\}$.

| Parameter | Dataset | Value |
|---|---|---|
| Annealing steps | Bikeshare | $10^5$ |
| | Wine | $10^5$ |
| | MIMIC 2 | $5 \times 10^4$ |
| | MIMIC 3 | $5 \times 10^4$ |
| | Mice protein | $5 \times 10^4$ |
| | Microsoft | $10^5$ |
| Dropout fraction | Bikeshare | 0 |
| | Wine | 0.1 |
| | MIMIC 2 | 0.3 |
| | MIMIC 3 | 0.5 |
| | Mice protein | 0.1 |
| | Microsoft | 0.1 |

Table 3: Dataset-specific parameters for the Distributed IB experiments.

## D  CITATION DIVERSITY STATEMENT

Science is a human endeavour and consequently vulnerable to many forms of bias; the responsible scientist identifies and mitigates such bias wherever possible. Meta-analyses of research in multiple fields have measured significant bias in how research works are cited, to the detriment of scholars in minority groups (Maliniak et al., 2013; Caplar et al., 2017; Chakravartty et al., 2018; Dion et al., 2018; Dworkin et al., 2020b). We use this space to amplify studies, perspectives, and tools that we found influential during the execution of this research (Zurn et al., 2020; Dworkin et al., 2020a; Zhou et al., 2020; Budrikis, 2020). We sought to proactively consider choosing references that reflect the diversity of the field in thought, form of contribution, gender, race, ethnicity, and other factors. The gender balance of papers cited within this work was quantified using a combination of automated gender-api.com estimation and manual gender determination from authors' publicly available pronouns. By this measure (and excluding self-citations to the first and last authors of our current paper), our references contain 5% woman(first)/woman(last), 12% man/woman, 14% woman/man, and 69% man/man. This method is limited in that a) names, pronouns, and social media profiles used to construct the databases may not, in every case, be indicative of gender identity and b) it cannot account for intersex, non-binary, or transgender people. We look forward to future work that could help us to better understand how to support equitable practices in science.

