# OpenReview forum: "Interpretability with full complexity by constraining feature information"
_ICLR.cc/2023/Conference — ICLR 2023 poster_

### Official Review · Reviewer_qhwf · 2022-10-21

**Confidence:** 4
**Correctness:** 3
**Technical Novelty And Significance:** 2
**Empirical Novelty And Significance:** 2
**Recommendation:** 6

**Clarity, Quality, Novelty And Reproducibility:**

Clarity: In general, this work is well-presented. The method and experimental setup are well-described.
Novelty: The information bottleneck has become an established tool in Deep Learning (see https://github.com/ZIYU-DEEP/Awesome-Information-Bottleneck for a long list of works). In particular, its properties have been used for to devise attribution and feature selection methods for interpretable ML before, for instance by Chen et al., Schulz et al., and Koh et al.
While the devised method can be seen as a global explanation technique and the others are local, they are still very similar in spirit. I have not seen the confusion matrix plots before and the quasi-information plots before, but to me it seems to be a combination of existing techniques.
The work is also highly similar to Murphy et al., where a) the distributed bottleneck is already proposed and b) the feature-wise information curves are presented.

Jianbo Chen, Le Song, Martin Wainwright, Michael Jordan: Learning to Explain: An Information-Theoretic Perspective on Model Interpretation. ICML 2018
Karl Schulz, Leon Sixt, Federico Tombari, Tim Landgraf: Restricting the Flow: Information Bottlenecks for Attribution. ICLR 2020
Pang Wei Koh, Thao Nguyen, Yew Siang Tang, Stephen Mussmann, Emma Pierson, Been Kim, Percy Liang: Concept Bottleneck Models. ICML 2020
Kieran A. Murphy, Dani S. Bassett: The Distributed Information Bottleneck reveals the explanatory structure of complex systems, arXiv 2204.07576

Reproducibility. I have no reproducibility concerns.


**Strength And Weaknesses:**

Strengths:

1. Overall, I found the presentation well-structured and clear.
2. Interpretability of AI algorithms is an important and timely topic. Specifically, the tradeoff between model complexity and interpretability is an important subject of study.
3. The presented techniques can provide reasonable insights into feature usage and feature selection. I think the visualizations, as shown in Figures 4 and 5, provide some useful insights into feature importance and can be useful for feature selection.
4. Many information-theoretic notions, such as entropy or mutual information, are sometimes hard to grasp. The distinguishability matrices used in this method help to better understand which aspects are important.

Weaknsses:

1. For the proposed method, an additional compression model needs to be trained, and this combination of models also needs to be trained for a sufficiently high number of betas. This is a lot more than required by other post-hoc techniques and may limit the usability of the method in practice.
2. While the method can quantify the information represented by the features, it does not give insights into how the information is used. For some applications, it might also be beneficial to see the effect of the information on the prediction. It would also be useful to analyze, whether there could be a faulty model that uses the right information but leads to wrong conclusions. Such behavior could not be discerned by the proposed method. Also, interactions are not visualized at all.
3. The motivation and the advantages of the proposed framework over other black-box explanation methods need to be improved. I am not exactly sure what the advantage of the proposed framework over other feature selection / attribution frameworks is. There are other methods that compute attributions, which also allow for more complex dependencies than GAMs or linear models. An advantage of GAMs / linear models is that these models are white box models. However, the proposed method is a black-box method. In this class a variety of methods including general constructs such as SHAP or the information bottleneck by Schulz et al. can also be applied with interactions between features.
4. So far, the method is restricted to categorical tabular data. In the paper, only categorical features are discussed. It would be nice to generalize and deploy the framework for continuous features and other data types, such as image or text data.


Minor comments:

1. Section 2 (Related work): The paragraph that discusses CCA, ANOVA and how linear correlation can be replaced by MI for feature selection seems very far away from the present work. Instead, the space could be used better to discuss works (e.g., those mentioned in the next box) that deploy the IB for interpretability.
2. Section 3 is very long. I would have appreciated subsections to better organize the content.
3. Sum H(X) and Sum I(X,U) (in Figure 2a) only denote the total (mutual) information in all features when the features are independent. Otherwise, it is only an upper bound. As far as I see independence was never assumed, so I was a bit confused here.

Questions:

1. Is the sparsity enforced (that the KLD for some features is 0 bits for some features, BikeShare in particular) or is it a side-product of the IB approach?
2. How long is the computational time of the method compared to other approaches? How many values for the KLD were used to compute the curves shown in Fig. 4?



**Summary Of The Paper:**

The paper proposes a framework for interpretability based on a feature-wise Information Bottleneck. It allows training classifiers that use optimal feature transformations maintaining only the required information. Two forms of visualizations of this information content are presented as global model explanations: Quasi-information plane trajectories with feature-wise information curves and confusion matrices for a single feature to show which information is used by the model. The interpretations are discussed for three tabular datasets with categorical features.

**Summary Of The Review:**

The research questions addressed in the paper are timely and interesting to the XAI community. The work presents some promising results and interesting visualizations for the information accessed by a model. However, its computational overhead, the close relation to other works that use IB, and unclear benefits over these works limit the possible impact, which is why I am currently leaning towards rejection.

---

> ### Author Response · Authors · 2022-11-12
> **Clarifications surrounding novelty, utility, and overhead**
>
> Thank you for your thorough review of the paper.  We summarized our modifications in a general comment and respond below to your specific comments:
>
> *> For the proposed method, an additional compression model needs to be trained, and this combination of models also needs to be trained for a sufficiently high number of betas. This is a lot more than required by other post-hoc techniques and may limit the usability of the method in practice.*
>
> This is not correct.  The model needs to be trained only once, and beta is swept over the course of training.  Each subplot in Figure 4 required only one training run.  We have added text in the Methods section to clarify this point.
>
> *> While the method can quantify the information represented by the features, it does not give insights into how the information is used. For some applications, it might also be beneficial to see the effect of the information on the prediction. It would also be useful to analyze, whether there could be a faulty model that uses the right information but leads to wrong conclusions. Such behavior could not be discerned by the proposed method. Also, interactions are not visualized at all.*
>
> If you want access to feature interactions and feature effects, there are intrinsically interpretable methods such as GAMs.  If you want to use full-complexity deep learning, there are simply no other intrinsically interpretable methods besides the Distributed IB.  We are proposing a route to interpretability where you don’t face any restrictions on the complexity of feature interactions, and the sacrifice is knowledge of those feature interactions.  Rather than seeing this as a bug, we see it as a feature -- this is the tradeoff that allows access to the unrestricted model space.
>
> *> The motivation and the advantages of the proposed framework over other black-box explanation methods need to be improved. I am not exactly sure what the advantage of the proposed framework over other feature selection / attribution frameworks is. There are other methods that compute attributions, which also allow for more complex dependencies than GAMs or linear models. An advantage of GAMs / linear models is that these models are white box models. However, the proposed method is a black-box method. In this class a variety of methods including general constructs such as SHAP or the information bottleneck by Schulz et al. can also be applied with interactions between features.*
>
> Our work sits between feature selection and white-box models that provide a window to the feature interactions and effects.  We compare closely to one method from each camp -- LassoNet for feature selection, and NAM for white-box models.  The Distributed IB can extract the specific partial information from each feature, and visualize exactly what that information is, before passing it along to an arbitrarily complex model that is trained as part of the full stack.
>
> The proposed method is distinct from feature attribution methods (such as the nice work by Schulz et al.), which are *post-hoc*, meaning they attempt to explain a trained model.  Instead, the Distributed IB focuses on the data itself, and trains models that utilize the most relevant information contained in the input features.
>
> We have clarified the Related Work around this point.
>
> *> So far, the method is restricted to categorical tabular data. In the paper, only categorical features are discussed. It would be nice to generalize and deploy the framework for continuous features and other data types, such as image or text data.*
>
> This is incorrect.  Many of the features in the datasets we considered are continuous-valued.  There is no such restriction on the features: anything that can be encoded can be bottlenecked.  We restricted our focus to tabular datasets in this work to establish the method as a route to  interpretability, though we see no limitations on applying the framework to other data modalities.
>
> *> Section 2 (Related work): The paragraph that discusses CCA, ANOVA and how linear correlation can be replaced by MI for feature selection seems very far away from the present work. Instead, the space could be used better to discuss works (e.g., those mentioned in the next box) that deploy the IB for interpretability.*
>
> We respectfully disagree.  The Painsky et al (2018) work generalizing CCA to use mutual information has deep connections with the information bottleneck and our work.  We have added to the Related Work some of the literature you recommend, but we feel the statistical analysis side of variable importance is an important point of comparison for many.

---

> > ### Author Response · Authors · 2022-11-12
> > **(continued)**
> >
> > *> Section 3 is very long. I would have appreciated subsections to better organize the content.*
> >
> > We thank the reviewer for this suggestion, and have reorganized the Methods into subsections for more clarity.
> >
> > *> Sum H(X) and Sum I(X,U) (in Figure 2a) only denote the total (mutual) information in all features when the features are independent. Otherwise, it is only an upper bound. As far as I see independence was never assumed, so I was a bit confused here.*
> >
> > The features X_i are encoded **independently** of one another, meaning that when every feature has been encoded maximally, the sum total of the encoded information will be the sum of their individual entropies.  We do not need to assume anything about the features’ independence for this to be true.
> >
> > *> Is the sparsity enforced (that the KLD for some features is 0 bits for some features, BikeShare in particular) or is it a side-product of the IB approach?*
> >
> > This is a side-product of the Distributed IB approach!
> >
> > *> How long is the computational time of the method compared to other approaches?*
> >
> > Details about the computational time were included in Appendix B:
> > “All Distributed IB experiments were run on a single computer with a 12 GB GeForce RTX 3060 GPU; 100k steps on the Bikeshare dataset takes a couple minutes, and on Microsoft about 2 hours.”
> >
> > We found LassoNet to take a little more time to run, and NAM significantly more when factoring in that you need to train an ensemble of ~25 models.
> >
> > *> How many values for the KLD were used to compute the curves shown in Fig. 4?*
> >
> > To be clear, only one training run was needed for each subplot in Fig. 4.  The bottleneck parameter $\beta$ was ramped for 100k training steps, from 2*10^-5 to 2 (specs listed in Appendix Table 2 and Table 3.  One sweep of $\beta$ yields all of the information allocations, at every D_KL.  We added some text to the Methods section to reinforce this important point.
> >
> > *> Novelty: The information bottleneck has become an established tool in Deep Learning (see https://github.com/ZIYU-DEEP/Awesome-Information-Bottleneck for a long list of works). In particular, its properties have been used for to devise attribution and feature selection methods for interpretable ML before, for instance by Chen et al., Schulz et al., and Koh et al. While the devised method can be seen as a global explanation technique and the others are local, they are still very similar in spirit. I have not seen the confusion matrix plots before and the quasi-information plots before, but to me it seems to be a combination of existing techniques. The work is also highly similar to Murphy et al., where a) the distributed bottleneck is already proposed and b) the feature-wise information curves are presented.*
> >
> > Indeed, the information bottleneck is a powerful tool for selectively extracting information.  However, distributing the bottlenecks to features before integration for the purpose of interpretability is, to the best of our knowledge, completely novel and absolutely essential for monitoring the flow of information.  Murphy et al. proposed the idea earlier this year but it required significant development.  The current work develops the necessary framework to use the Distributed IB for interpretable machine learning.

---

> > > ### Comment · Reviewer_qhwf · 2022-11-17
> > > **Thank you for the detailed feedback!**
> > >
> > > I thank the authors for their detailed feedback and clarifications. The novelty of the approach is still questionable. Nevertheless, based on the improvements and clarifications added to the paper, I will raise the score accordingly.

---

### Official Review · Reviewer_zAhj · 2022-10-24

**Confidence:** 4
**Correctness:** 4
**Technical Novelty And Significance:** 3
**Empirical Novelty And Significance:** 3
**Recommendation:** 8

**Clarity, Quality, Novelty And Reproducibility:**

The paper is clear and proposes a novel variation of the Distributed Information Bottleneck framework.

**Strength And Weaknesses:**

Good:
The approach goes beyond selecting a (discrete) number of active features but rather continuously varies the degree of compression.
An in depth analysis can be performed by looking at a confusion matrix type of representation that is in principle richer.

Bad:
The claim that the approach improves the interpretability of complex models needs to be better defended: the type of analysis offered in the end is often unclear or complicated:
- capacity to extract relevant features: the method allows to quantify <<how much information is required - total, and from each feature - for a given level of predictability>>, ok how is this different from the information that we can get from the Lasso Path along the regularization parameter using the LARS algorithm? How does one use this information in practice?
- as an example of analysis, the author report a confusion matrix type of representation that allows to conclude that the model is first paying attention to discriminating specific ranges of values for some variables. How different would this be from partitioning in a few bins each variable and then run a recursive feature elimination selection technique to identify which (interacting) ranges of which variables are important?
- <<The capacity for successive approximation, allowing detail to be incorporated gradually >> how can this in practice be used to offer insights? is it possible to identify some prototypical scenarios, i.e. if this happens than this can be concluded?
- the experimental section would benefit from including some artificial cases where the known important interactions are known a-priori and it can be shown that they are recoverable by the proposed approach but not by other baselines (such as Lasso or recursive feature elimination selection techniques).

**Summary Of The Paper:**

The authors develop an interpretability framework based on information theory. The advantage/novelty of the approach being that it works directly on the 'informativeness' of the individual features and their joint interactions w.r.t. a given target under potentially a variety of losses and a variety of neural network architectures as predictive models.

**Summary Of The Review:**

The ideas presented are of interest but more effort should be put in demonstrating the capacity of the framework to offer useful (and understandable) insights.

After the authors' modifications I feel the paper has improved and is ready for publication.

---

> ### Author Response · Authors · 2022-11-12
> **More support in manuscript that the proposed method improves interpretability**
>
> Thank you for your sincere engagement with our paper.  We summarized our modifications in a general comment and respond below to your specific comments:
>
> *> Capacity to extract relevant features: the method allows to quantify <<how much information is required - total, and from each feature - for a given level of predictability>>, ok how is this different from the information that we can get from the Lasso Path along the regularization parameter using the LARS algorithm? How does one use this information in practice?*
>
> LARS and general feature selection methods face a fundamental constraint in the form of hard inclusion/exclusion of features.  The Distributed IB learns a compression of each feature and can extract only the relevant information necessary for different values of the bottleneck parameter.  The difference allows for two enormous advantages upon which we focus our experimental results in the paper.  1) The path of incorporating more information into the model becomes continuous, and at each step there will be continuous proportions of information from the various features.  2) The important information contained within the features can be inspected as its own significant source of interpretability, and there is simply no analogue in other feature selection methods.
>
> *> As an example of analysis, the author report a confusion matrix type of representation that allows to conclude that the model is first paying attention to discriminating specific ranges of values for some variables. How different would this be from partitioning in a few bins each variable and then run a recursive feature elimination selection technique to identify which (interacting) ranges of which variables are important?*
>
> The scheme proposed by the reviewer should achieve the same ends as the Distributed IB, though with much more work and injected arbitrariness.  The Distributed IB requires only a single training run with the features input wholesale, sweeping through strengths of the bottleneck and effectively learning its own partitioning of every feature for every information rate.
>
> *> <<The capacity for successive approximation, allowing detail to be incorporated gradually >> how can this in practice be used to offer insights? is it possible to identify some prototypical scenarios, i.e. if this happens then this can be concluded?*
>
> Perhaps some clarity will come by viewing the revised Fig. 5, where we visualize several approximations for the Bikeshare dataset (regressing the number of bikes rented in Washington, D.C. based on time and weather features) instead of the less intuitive medical dataset MIMIC-II. The first approximations we find, using only two and four bits of information about the input features, are quite comprehensible.  The time of day is chunked into roughly three categories: commuting period, middle of the night, and everything else.  The apparent temperature is roughly divided into hot and cold distinctions.
>
> If we pause at this point, we have a model that uses information about only two or three features, and it performs as well as the linear models we tested.  If we continue adding information, we see the next couple of bits significantly improve performance, and go to refining the time of day, the season, etc.  Eventually we reach a point where there is too much information to be very interpretable, but the strength of the Distributed IB is we inherited a control knob to move along that continuum at will.  We were able to add more detail after comprehending the information allocation of a simpler allocation.  The ability to gradually increase the complexity of the model to comprehend is, we think, a valuable route to insights.
>
> *> The experimental section would benefit from including some artificial cases where the known important interactions are known a-priori and it can be shown that they are recoverable by the proposed approach but not by other baselines (such as Lasso or recursive feature elimination selection techniques).*
>
> We thank the reviewer for this suggestion.  We are working on an artificial scenario to include by the end of the discussion period.

---

### Official Review · Reviewer_rdFY · 2022-10-25

**Confidence:** 3
**Correctness:** 2
**Technical Novelty And Significance:** 2
**Empirical Novelty And Significance:** 3
**Recommendation:** 6

**Clarity, Quality, Novelty And Reproducibility:**

The language and presentation needs to be improved as already mentioned above. The algorithm is novel, but the merits are not clear.

**Strength And Weaknesses:**

1. Overall, the paper needs to make the language clearer and improve the presentation of the algorithm.

Examples of text that are unclear:

    a. Abstract: "The central object of analysis is not a single trained model, but rather a spectrum of models serving as approximations that leverage variable amounts of information and range from the uninformative to the most performant model obtainable."

    b. Abstract: "tackling the problem of feature selection with inclusion/exclusion existing on a learned continuum."

    c. The paper keeps referring to 'a spectrum of models'. But it is not clear what that means in this context. Please provide a clear definition of it in the beginning.

    d. Section 4.1: "The signal obtained by the Distributed IB can help decompose through datasets with a large number of features and complex interaction effects."

    e. Section 3: "The Distributed IB (Estella Aguerri & Zaidi, 2018) concerns..." -- there is some confusion. Distributed IB is a reference to a previous work as well as being used as the name of the proposed algorithm (Table 1). Should disambiguate in text.


2. Two of current most popular techniques for interpretability LIME [r1] and SHAP [r2] are missing in discussions and experiments.


3. In the experiments section, there is no comparison with existing techniques for interpretability.


4. Section 3 should present the techniques for training and information plane visualization in Algorithm formats. A schema of the overall architecture of the algorithm should also be illustrated as a picture for clarity.


5. It is hard to understand whether the proposed technique is a 'global' interpretability technique or 'local'. As per my understanding from statements like in Section 3 "...The penalty has the effect ... information that is least helpful ... is discarded.", the technique is 'local'.

A 'global' (e.g., LASSO) technique allows us to identify the most important features/feature values across all training data.

A 'local' (e.g., LIME [r1]) technique allows us to identify the features/feature values that lead to a specific prediction for a specific input data.


References

[r1] Ribeiro, Marco Tulio, Sameer Singh, and Carlos Guestrin. "Why should I trust you?: Explaining the predictions of any classifier." Proceedings of the 22nd ACM SIGKDD international conference on knowledge discovery and data mining. ACM (2016).

[r2] Lundberg, Scott M., and Su-In Lee. "A unified approach to interpreting model predictions." Advances in Neural Information Processing Systems (2017).

>>>>>>>>>>> Update after author response

I thank the authors for responding to my comments. At this time I do not have sufficient reasons to change my scores. Some comments:

1. "...from the uninformative to the most performant model obtainable..." -- The range 'from uninformative .... to ... performant' makes little sense (like saying 'from brightest ... to hottest'). It is not clear what the two ends of the scale 'uninformative' and 'performant' mean i.e., are these in terms of interpretability or accuracy?

2. Thanks for pointing out that the method is 'global'. Global methods are not as interesting (w.r.t research and application) as 'local' methods that explain model response on specific examples, especially when the number of training examples is large and the number of features is small. When the number of features is large (few hundreds), the global feature selection methods are useful, but usually the objective is then to clean the data rather than to enhance interpretability.

3. Figure 4 needs additional labels/captions on the figures to improve readability. E.g., Distributed IB/LassoNet needs to be displayed on the plots, else it is difficult to follow the dense caption text.

4. I am still of the opinion that the language needs to be made clearer and straight-forward, figures should be better presented, and the 'interpretability' aspect is not convincing -- for a 'global' interpretable algorithm, it outputs too much information and requires too much effort from an end user.



**Summary Of The Paper:**

The paper presents a technique to improve interpretability of models by compressing the information in features and feature values using Distributed Information Bottleneck. The redundant and useless features/feature values are intended to be removed by the bottleneck layer so that only the remaining relevant features/values are used for model prediction. This promotes model sparsity which makes the predictions more comprehensible.

**Summary Of The Review:**

While the application of the information bottleneck is novel, relevant literature has been missed out and comparison with existing techniques for interpretability needs to be covered in the experiments.

---

> ### Author Response · Authors · 2022-11-12
> **The gained interpretability is global, not local  |  Clarified text, contextualization in related work, and algorithm presentation**
>
> Thank you for your thorough review of the paper.  We summarized our modifications in a general comment and respond below to your specific comments:
>
> *> Examples of text that are unclear:
> Abstract: ”The central object of analysis is not a single trained model, but rather a spectrum of models serving as approximations that leverage variable amounts of information and range from the uninformative to the most performant model obtainable.”*
>
> Intrinsic interpretability methods (e.g. NAM) generally focus on a single trained model as the object to interpret, while feature selection methods (e.g. LassoNet) generally focus on a discrete spectrum of models that utilize different subsets of features.  The proposed methodology finds a continuous spectrum of models, and interpretability is found by analyzing how information from the features is gradually incorporated.  When there is no information about the input features used by the model, it is uninformative: every input is mapped to the same output.  At the other side of the spectrum, all feature information is utilized in the most performant model obtainable.
>
> *> Abstract: ”tackling the problem of feature selection with inclusion/exclusion existing on a learned continuum.”*
>
> Feature selection is nearly always discrete: is feature A included or excluded from the subset of most informative features?  The Distributed IB offers a powerful alternative whereby information from the features can be continuously ramped to zero; this allows for smooth optimization and a source of additional interpretability in the form of a prioritization of the most important information *within* each feature.  The continuum of feature inclusion is learned through optimization, which is what we mean by ‘learned continuum’.
>
> *> The paper keeps referring to 'a spectrum of models'. But it is not clear what that means in this context. Please provide a clear definition of it in the beginning.*
>
> Every value of the bottleneck parameter \beta corresponds to a model utilizing a different amount of information about the input features.  By sweeping \beta, we traverse a spectrum of models.  We have revised the text in the Methods section for clarity around this point of confusion, and revised Figure 2 to de-emphasize trajectories and instead focus on regions of possible entropy allocations/regions of possible models.  The Distributed IB probes the boundary of these possibility regions, which is a spectrum/continuum of models.
>
> *> Section 4.1: "The signal obtained by the Distributed IB can help decompose through datasets with a large number of features and complex interaction effects."*
>
> We have revised this portion of the text for clarity.
>
> *> Section 3: "The Distributed IB (Estella Aguerri & Zaidi, 2018) concerns..." -- there is some confusion. Distributed IB is a reference to a previous work as well as being used as the name of the proposed algorithm (Table 1). Should disambiguate in text.*
>
> We have not changed the Distributed IB algorithm from its original use in relation to the problem of multi-terminal source coding, so we preserved the name.  Our primary contribution is to apply the Distributed IB algorithm to inputs of a black-box model, and to build a framework to recover interpretable insight from the Distributed IB optimization.
>
> *> Two of current most popular techniques for interpretability LIME [r1] and SHAP [r2] are missing in discussions and experiments.*
>
> *> It is hard to understand whether the proposed technique is a 'global' interpretability technique or 'local'. As per my understanding from statements like in Section 3 "...The penalty has the effect ... information that is least helpful ... is discarded.", the technique is 'local'.*
>
> The proposed methodology is a **global** interpretability technique, which is why we do not compare to local interpretability techniques like LIME and SHAP.  The important information across the entire space of inputs is found at once; interpretability is not built example by example.  We have added text to the Related Work to make this more clear.

---

> > ### Author Response · Authors · 2022-11-12
> > **(continued)**
> >
> >
> > *> In the experiments section, there is no comparison with existing techniques for interpretability.*
> >
> > We think the reviewer is mistaken, or might expect comparison to local interpretability techniques that are irrelevant to the current study (see above response).  In Figures 4 and 5 we show side-by-side comparisons with existing techniques for interpretability in order to illustrate the capabilities and compromises when using the Distributed IB.
> >
> > In Figure 4 we compare with LassoNet, a method that reveals feature importance by training with an L1 loss on feature inclusion/exclusion into a model.  In contrast to the discrete set of models found by LassoNet---each utilizing a different number of input features---the Distributed IB finds a far richer continuum of models that reveal how much information comes from each feature for different levels of predictive power.
> >
> > In Figure 5 we compare with Neural Additive Models (NAM), which reveals feature effects at the expense of higher order feature interactions.  We display the feature effects found by NAM next to the confusion matrices for each learned feature compression scheme.
> >
> > In both sets of experiments, we have provided a direct comparison of the interpretability signal found by optimizing the Distributed IB with that of a relevant interpretability technique.
> >
> > *> Section 3 should present the techniques for training and information plane visualization in Algorithm formats. A schema of the overall architecture of the algorithm should also be illustrated as a picture for clarity.*
> >
> > We will add Algorithm formats for the Distributed IB optimization to the Methods section by the end of the discussion period.

---

> > > ### Comment · Reviewer_rdFY · 2022-11-22
> > > **Thanks for the additional effort**
> > >
> > > I am increasing my score in light of the effort the authors have spent in improving clarity of the paper.

---

> ### Author Response · Authors · 2022-11-17
> **Entirely new, practical, and useful tool within interpretable ML**
>
> We greatly appreciate the reviewer’s engagement and directness during this discussion period.  To match the reviewer’s directness of comments: we feel strongly that the proposed method of **tracking information usage by full complexity models, for all levels of approximation to a relationship in data** represents an entirely new, practical, and useful tool within interpretable ML.
>
> There is unfortunately a large discrepancy in the value that we see in the proposed method compared to how the reviewer sees it.  If we cannot bridge the discrepancy for this submission, we are appreciative to at least have a better understanding for a future submission.
>
> *2. Thanks for pointing out that the method is 'global'. Global methods are not as interesting (w.r.t research and application) as 'local' methods that explain model response on specific examples, especially when the number of training examples is large and the number of features is small. When the number of features is large (few hundreds), the global feature selection methods are useful, but usually the objective is then to clean the data rather than to enhance interpretability.*
>
> *4. I am still of the opinion that the language needs to be made clearer and straight-forward, figures should be better presented, and the 'interpretability' aspect is not convincing -- for a 'global' interpretable algorithm, it outputs too much information and requires too much effort from an end user.*
>
> We are surprised by the perspective that global methods are not as interesting as local methods of interpretability, and that our algorithm outputs too much information and requires too much effort.
>
> If you will humor us briefly, we would be grateful if you would look at Figure 9, at the end of the pdf, showing every feature’s compression scheme at 13 levels of approximation on the Bikeshare dataset.  Look at how much can be learned about the relationship between the features and the output by seeing in which **features** and in which **feature values** the most predictive information resides!  Distinguishing commuting times from the middle of the night, and then from everything else is the most important information in the data.  Eventually winter is worth distinguishing from the three warmer seasons, the top 10% of humidity values is worth distinguishing from the rest, and so on.  Only 5 bits is necessary to beat the linear models, and we can see exactly where those 5 bits of information come from!
>
> Everything in this plot came out of a single training run (“too much effort”), and did not require building up interpretability example by example (in contrast to a local method).  Local methods are vulnerable to over-interpretation, whereby the user pieces together an understanding from examples; global methods lay everything out at once.  We recognize the value of both strategies to interpreting machine learning models, but we do not share the view that “global methods are not as interesting (w.r.t. research and applications) as local methods”.
>
> Regarding the remaining comments, we have uploaded a revision of the paper and responded below:
>
> *1. "...from the uninformative to the most performant model obtainable..." -- The range 'from uninformative .... to ... performant' makes little sense (like saying 'from brightest ... to hottest'). It is not clear what the two ends of the scale 'uninformative' and 'performant' mean i.e., are these in terms of interpretability or accuracy?*
>
> Now we understand the source of confusion; thank you for the example.  The range of models has the two endpoints shown in Fig. 2b, and what we meant by “uninformative” is that the model at the red dot completely ignores the inputs -- it outputs the same thing regardless of the input.  “Uninformative” is unclear, especially given the context of interpretability.  We have removed this language for lack of a clear alternative.
>
> *3. Figure 4 needs additional labels/captions on the figures to improve readability. E.g., Distributed IB/LassoNet needs to be displayed on the plots, else it is difficult to follow the dense caption text.*
>
> We thank the reviewer for the helpful suggestions.  We have added the suggested headings as well as additional labeling of the highlighted features in each plot.  We hope the resulting plot is clearer.

---

> > ### Comment · Reviewer_rdFY · 2022-11-18
> > **Paper needs more work**
> >
> > I still maintain that the paper needs to present the results in a much more intuitive manner. For a paper that targets 'interpretability', the presentation is surprisingly convoluted. As already pointed out, the figures are hard to understand, even Figure 9. The point of interpretability is to reduce the cognitive burden on the end user. The paper does not do that and instead throws up the entire data all at once and hopes that the end user will make sense of it. In that aspect it acts just as an ML model like any other.
> >
> > Suggestion 1: first try to state what the most important part is (w.r.t interpretability) in simple english and output that explanation in a structured manner that might be actionable by an automated process for a downstream task. For this, you could fix a particular bit size -- do a little research into the explainability research such as (Molnar 2022) -- I think 3 bits might be a good number since humans probably are more comfortable at that level of complexity. You might alternatively, select a %RMSE (relative to full-model RMSE) threshold by an *automated* means and decide the bit value based on that. Next, for a *number* of datasets, explain the results (keeping bits/%RMSE fixed) with the structured output as suggested in Suggestion 1. You should defer the compression matrices to the Appendix -- they for most part interrupt the story and it does not help that the figures are poorly labeled and densely captioned.
> >
> > Suggestion 2: Use more datasets and competing algorithms. For this, you could refer to Lemhadri 2021 for experiment/methodology -- specifically, a plot like the Figure 5 of Lemhadri 2021 would be useful for each dataset. That way, a more unbiased comparison can be done with an existing literature.
> >
> > Suggestion 3: it would be more interesting to extend the algorithm to get local explanations. In other words, could there be a model that would learn how much to throttle information from one input feature given the values of other input features?
> >
> > I believe an implementation of Suggestion 1 and to some extent Suggestion 2 would be very useful and could improve the scores for the paper.

---

> > > ### Author Response · Authors · 2022-11-19
> > > **Concluding thoughts**
> > >
> > > We thank the reviewer for offering thoughtful suggestions.  As the primary suggestion (#1) is well-contained within the scope of the current manuscript, we do not think it necessary.  However, we recognize the wisdom in breaking a story into more digestible pieces, as there is a lot going on with monitoring information from features and feature values, for all levels of approximation.
> > >
> > > We would like to clarify that there is a large difference between presenting a signal that can be comprehended gradually -- even if it is hard to grasp when one is not used to thinking from an information theory perspective -- and 'an ML model like any other.'  The primary reason ML is not by default interpretable is the signal cannot be modularized and comprehended in piecemeal fashion.  We have presented a method that allows sequential addition of detail, and the richness of the signal is (to us) a feature, not a bug.

---

### Official Review · Reviewer_sKPb · 2022-10-27

**Confidence:** 3
**Correctness:** 3
**Technical Novelty And Significance:** 3
**Empirical Novelty And Significance:** Not applicable
**Recommendation:** 6

**Clarity, Quality, Novelty And Reproducibility:**

Clarity/Quality-- Medium:
The text is clear but the figures are very difficult to understand.

Novelty--High:
The paper introduces a new class of interpretable models.

Reproducibility-- Medium:
The code for the experiments was included but it is not well documented i.e there were no instructions on how to run it or which functions to call..etc

**Strength And Weaknesses:**

Strength:
---

- The idea is quite novel, constraining the amount of information going into model for interpretability is very interesting.
- The idea is well motivated.
- Unlike GAMs the approach allows feature interaction since there are no restrictions on how the features can interact after IB.
- Using such an approach on tabular data does not cause model accuracy degradation.


Weakness:
---
- I am concerned about the interpretability from the following aspects:
     - Is this model actually interpretable since we are not sure how features interact?
     - What does it mean (in terms of interpretability) to need part of a feature? Like using some information about the age of humans important?
    - Explanations produced by the model are very hard to interpret, I read the paper multiple times and I don't fully understand the interpretations produced in figure 5.

- The paper is well written but the figures are very difficult to understand, better interpretations should be included,
- Given that this paper is mainly about interpretability more interpretability results are needed, for example, the use of a synthetic dataset showing that the optimal amount of feature information is actually extracted by IB. A user study can also be useful.
- The proposed model only applies to tabular datasets and can not be used in other domains like images, language and time series (although this is the case for NAM).



**Summary Of The Paper:**

The paper proposes a new class of interpretable models, instead of limiting model complexity (like using simple interpretable models) or limiting the feature interaction (like in GAM), the paper constrains the amount of information in the model by using Distributed Information Bottleneck IB. IB introduces a constraint after every feature is processed and before any interaction effects between the features can be incorporated. The features are compressed to optimal representations which communicate the most relevant information to the rest of the model This can be viewed as a penalty for the amount of information used by a model.

**Summary Of The Review:**

The overall idea is very interesting. However, I am not completely convinced that the proposed models are in fact interpretable since we do not know how features actually interact. Also, the interpretations provided by the paper were very hard to understand which again questions the overall interpretability of the method.  Interpretations are generally created for humans to have a better understanding of the models and how it uses different features. I am not sure this model achieves this.

---

> ### Author Response · Authors · 2022-11-12
> **Interpretability comes through revealing the most important information in features  |  Improved figures and added more examples**
>
> Thank you for your thoughtful questions and thorough review of the paper.  We summarized our modifications in a general comment and respond below to your specific comments:
>
> *> Is this model actually interpretable since we are not sure how features interact?*
>
> While revealing feature effects and feature interactions is an intuitive and well-worn strategy for interpretability, we propose that revealing what information is being used by a black-box model can serve as a powerful new source of interpretability.
>
> The utility is ultimately determined in practice: take the revised Figure 5 tracking the information in the Bikeshare dataset (we moved the MIMIC-II figure to the Appendices; readers will presumably have more intuition about how weather affects bike rentals).  Linear models perform poorly (Table 1) so we would like to train a full complexity model while retaining a sense of how the model operates.
>
> What interpretability is gained by tracking the information with the Distributed IB?  We discover that the time of day (the `hour` feature) contains the most predictive information, and further *what* the most predictive information is, to a first approximation: whether it’s commuting time or not.  With a little more information about temperature (whether it’s hotter than around 20C outside), the model performs about as well as the linear model NAM.  We continue incorporating information from the features in order of importance, finding (for example) that distinguishing winter from spring/summer/fall is the most important information in the `season` feature, that distinguishing weekends from weekdays is important, etc.  Without constraints on how the features interact, we improve dramatically in performance though the best we can do is monitor the information *before* it enters the model.
>
> Would you agree that we have gained a significant amount of interpretability about how the black-box model operates, even though we do not know the feature interactions?
>
> *> What does it mean (in terms of interpretability) to need part of a feature? Like using some information about the age of humans important?*
>
> One intuitive example of partial information about a feature is a simple threshold, like one would find in a decision tree.  Is the age of a person greater than 50?  Such a threshold reduces the full information of a feature to a single bit or less.  More generally, information may be thought of in terms of possible distinctions.  It may make little predictive difference whether age 43 and age 44 are distinguished, in which case a model trained with the information bottleneck penalty will clump the two feature values together.  On the other hand, there may be a marked change in output between age 30 and age 60, so that distinction is worth preserving for the model.  The important information---i.e., the important distinctions---can be a strong source of interpretability, particularly to someone with relevant domain knowledge, and this is what we show by the confusion matrices in Figure 5.
>
> *> Explanations produced by the model are very hard to interpret, I read the paper multiple times and I don't fully understand the interpretations produced in figure 5.*
>
> We have revised Figure 5 to use the more intuitive Bikeshare dataset as its focus, and to more clearly lay out the figure.  We have also clarified the discussion of the figures in the Experiments section.
>
> *> Given that this paper is mainly about interpretability more interpretability results are needed, for example, the use of a synthetic dataset showing that the optimal amount of feature information is actually extracted by IB. A user study can also be useful.*
>
> We will include an analysis of a synthetic dataset by the end of this discussion period.
>
> *> The proposed model only applies to tabular datasets and can not be used in other domains like images, language and time series (although this is the case for NAM).*
>
> In principle any input to a machine learning model can be bottlenecked, though this is outside the scope of the present work.  We restricted our focus to tabular datasets because interpretation is straightforward when the inputs have meaning.
>
> *> Reproducibility-- Medium: The code for the experiments was included but it is not well documented i.e there were no instructions on how to run it or which functions to call..etc*
>
> We have refactored the code to subclass `tf.keras.Model` and leverage the high level `keras` API (included in the re-uploaded supplementary files).  Training with the Distributed IB is nearly as easy as training any black-box model, and the information allocation by feature is recorded in the `history` object during training.  With additional callbacks the feature interactions can be saved over the course of training as well.  Details for usage have been added to Appendix B.

---

> > ### Comment · Reviewer_sKPb · 2022-11-21
> > **Thank you**
> >
> > Thank you for addressing my concerns based on this I increased my score.

---

### Author Response · Authors · 2022-11-12
**Novel source of interpretability  |  Revised for clarity and more support**

We thank the reviewers for the detailed and constructive feedback. We appreciate that the central idea of the work---to track the flow of information into a black-box model as a source of interpretability---was recognized as **novel** (R1, R2, R3), **timely/useful/of interest** (R3, R4), and **well-motivated** (R1).  Shortcomings revolved around our presentation of the method: the merits and utility of the acquired interpretability signal (R1, R2, R3), comparison/contextualization with related work (R2, R4), and the overhead of using the method in practice (R4).

As the reviewers’ concerns centered around presentation---whereas the core premise of the work was largely appreciated---we are hopeful that the manuscript revisions and the following clarifications will convincingly demonstrate the value of tracking information from features into a full-complexity machine learning model.

Before we respond to each reviewer individually, we would like to highlight the distinction of the proposed methodology relative to established strategies for interpretable ML.  Relative to local  interpretability methods like LIME or SHAP, Distributed IB provides **global interpretability**: instead of building toward comprehensibility through individual input-output examples, we find the most relevant information about the entire space of inputs at once.  Relative to feature selection and variable importance, the Distributed IB can extract only the partial information necessary from each feature and provides a far richer signal (see the comparison to LassoNet in Fig. 4).  Relative to nearly all other methods with intrinsic interpretability, which place constraints on feature interactions, the Distributed IB allows for unconstrained model complexity at the expense of knowledge of feature effects.  Instead, it reveals what information within each feature that is most important---a lesser signal than feature effects but the most for which we can hope in exchange for unfettered model complexity.  **Tracking the information flow into a model is an entirely new strategy for interpretable ML, and we have added clarification around this point in the revised manuscript.**

List of revisions to the manuscript:
- New Figure 5 (feature value confusion matrices), on a more intuitive dataset (Bikeshare) and with clarified presentation
- Multiple feature compression examples added to Appendices
- Edited Figure 2 to display possibility regions instead of example trajectories
- Related work: added global vs local, intrinsic vs post-hoc interpretability
- Reorganized Methods section for clarity
- Refactored code -- subclassing `tensorflow.keras.Model` to utilize the high level API of keras (`model.compile()`, `model.fit()`, etc.).  We attached a self-contained iPython notebook to run the refactored code on the “Wine” dataset
- We are working on a synthetic example to add by the end of this discussion period

---

### Author Response · Authors · 2022-11-19
**Added synthetic example for intuition about monitoring information used by a full complexity model**

In addition to the revisions listed in the previous comment, we have added a synthetic example to the Appendices (Figure 6)  that should help build intuition for our framework.  The example is a simple case with two interventions affecting a binary outcome variable, and everything (including the embeddings) can be visualized directly.  It was a helpful suggestion from Reviewers zAhj and skPb.

We again thank the reviewers for their engagement with our manuscript!

---

### Decision · Program_Chairs · 2023-01-20

**Decision:**

Accept: poster

**Justification For Why Not Higher Score:**

The reviewers are not unanimous in their assessments of novelty, clarity, impact. I do wonder whether the scores may reflect some amount of anchoring (reticence on reviewers to change their scores/impression too much) and so, had the paper been in its final form, it is conceivable it would have been received better.

**Justification For Why Not Lower Score:**

Three reviewers moved from reject to accept recommendations and so I think that reflects a strong signal that the work is viewed as worthy of the broader community's attention. The fourth review recommend a strong accept. I personally see an interesting new take on an important problem. It may not be the cleanest version of this idea, but I think the community needs new ideas like this one.

**Metareview: Summary, Strengths And Weaknesses:**

This paper proposes to track the information between features and model outputs and, through compression/distortion, identify a spectrum of models utilizing a variable amount of information. This spectrum of models then provides a global type of interpretability.

The reviews initially penalized the work for clarity. Through considerable efforts on the part of the authors and good engagement by the reviewers, these clarity issues were resolved to enough of an extent that three reviewers raised their scores to recommend acceptance. It seems that the work is recognized as novel and impactful---certainly interpretability is a key problem facing blackbox methods in ML.

Regard weaknesses, the reviewers are perhaps not entirely convinced that the paper is in its clearest form, but I think the authors have to be allowed some control.

**Note From Pc:**

if the above contains the word "oral" or "spotlight" please see: "oral" presentation means -> notable-top-5% and "spotlight" means -> notable-top-25%. As stated in our emails, we are disassociating presentation type from AC recommendations

**Summary Of Ac-Reviewer Meeting:**

N/A